# Altered sulfation status of FAM20C-dependent chondroitin sulfate is associated with osteosclerotic bone dysplasia

Toshiyasu Koike[1], Tadahisa Mikami [1], Jun-Ichi Tamura[2] & Hiroshi Kitagawa [1] ✉

Raine syndrome, a lethal osteosclerotic bone dysplasia in humans, is caused by loss-of-function mutations in *FAM20C*; however, *Fam20c* deficiency in mice does not recapitulate the human disorder, so the underlying pathoetiological mechanisms remain poorly understood. Here we show that FAM20C, in addition to the reported casein kinase activity, also fine-tunes the biosynthesis of chondroitin sulfate (CS) chains to impact bone homeostasis. Specifically, FAM20C with Raine-originated mutations loses the ability to interact with chondroitin 4-*O*-sulfotransferase-1, and is associated with reduced 4-sulfation/6-sulfation (4S/6S) ratio of CS chains and upregulated biomineralization in human osteosarcoma cells. By contrast, overexpressing chondroitin 6-*O*-sulfotransferase-1 reduces CS 4S/6S ratio, and induces osteoblast differentiation in vitro and higher bone mineral density in transgenic mice. Meanwhile, a potential xylose kinase activity of FAM20C does not impact CS 4S/6S ratio, and is not associated with Raine syndrome mutations. Our results thus implicate CS 4S/6S ratio imbalances caused by *FAM20C* mutations as a contributor of Raine syndrome etiology.

Skeletal bone development and homeostasis are maintained by reciprocal control of bone formation and resorption. A breakdown of this balance facilitates the onset of a wide range of bone disorders, including osteoporosis and osteosclerosis, which are often characterized by abnormal changes in bone mineral density (BMD). A lethal sclerosing bone dysplasia, Raine syndrome, is an autosomal recessive disorder caused by mutations in *FAM20C*, one of the three related proteins constituting a "family with sequence similarity 20, members A–C"[1–5]. Affected individuals show generalized osteosclerosis with periosteal bone formation[1–4], implying possible inhibitory roles of FAM20C in osteoanabolism or biomineralization. Intriguingly, however, FAM20C depletion in mice reportedly does not result in the development of osteosclerosis but rather manifests as bone defects and rachitic skeletal phenotypes[6–8]. Although some *FAM20C* mutations in non-lethal Raine syndrome patients are associated with fibroblast growth factor (FGF) 23-dependent hypophosphatemic rickets or osteomalacia[9,10], the mechanisms underlying

these contradictory findings in patients and mice models remain elusive.

FAM20C was originally identified as dentin matrix protein 4 (DMP4), a calcium-binding protein that modulates odontoblast differentiation[11]. Recent biochemical studies have revealed that *FAM20C* encodes a Golgi casein kinase that phosphorylates serine (Ser) residues of secreted extracellular proteins[12–15], such as small integrin-binding ligand *N*-linked glycoproteins (SIBLINGs), and a phosphate-regulating hormone, FGF23. As the phosphorylation status of SIBLINGs and FGF23 determines the positive or negative regulation of biomineralization, an imbalanced profile of FAM20C-dependent protein phosphorylation in vivo has been postulated as a plausible scenario accounting for species-specific phenotypic manifestations[4,13].

Both protein and glycan moieties of proteoglycans (PGs) are phosphorylated in secretory pathways; the corresponding glycan is a glycosaminoglycan (GAG)-protein linkage region tetrasaccharide that is covalently attached to specific Ser residues embedded in a panel of

[1]Laboratory of Biochemistry, Kobe Pharmaceutical University, Higashinada-Ku, Kobe 658-8558, Japan. [2]Department of Agricultural, Life and Environmental Sciences, Faculty of Agriculture, Tottori University, Tottori 680-8551, Japan. ✉e-mail: kitagawa@kobepharma-u.ac.jp

core proteins[16,17]. Two representative GAGs, chondroitin sulfate (CS) and heparan sulfate (HS), are universally distributed on the cell surface and in extra/pericellular matrices in the form of PGs. CSPGs and HSPGs form physiological milieus that support various cellular processes. The broad-spectrum actions of PGs are highly dependent on the structural characteristics of GAG chains, such as size, number of GAGs per core protein, and position and degree of sulfation[18–20]. CS and HS chains are linear polysaccharides of repeating disaccharide units composed of glucuronic acid (GlcA) and their respective amino sugars, N-acetylgalactosamine (GalNAc) and N-acetylglucosamine (GlcNAc). Since both sugar chains are synthesized on the linkage tetrasaccharide GlcAβ1–3 Galβ1–3 Galβ1–4Xyl, where Gal and Xyl represent galactose and xylose, respectively, strict biosynthetic regulation of the linkage tetrasaccharide is a prerequisite for the functional assembly of the CS and HS chains (Fig. 1a)[16,17]. Notably, transient phosphorylation and subsequent dephosphorylation of the Xyl residue frequently occur during linkage saccharide biosynthesis[21,22]. We previously demonstrated that FAM20B, another FAM20 family member, is a Golgi kinase specific for the Xyl residue of the linkage tetrasaccharide (Xyl kinase, XYLK)[23] and that the FAM20B-dependent phosphorylation status of the linkage region enables fine-tuning of GAG production via the cooperative actions of several glycosyltransferases and a Xyl phosphatase[24–27]. The evolutionary relationships of FAM20 family members suggest that mammalian FAM20B proteins are a direct ancestral branch and that FAM20C may arise as a result of duplication and evolution of the ancestral gene[5,28]. Consequently, although the reported substrate preferences of FAM20B and FAM20C are quite different[12–15,23,26,29], we hypothesized that FAM20C is also involved in GAG biosynthesis.

Here, we reveal the regulatory roles of FAM20C in the machinery for GAG biosynthesis as an additional XYLK, and, more importantly, for the synthesis of CS chains through its non-enzymatic ability to interact with a CS-specific sulfotransferase, chondroitin 4-O-sulfotransferase-1 (C4ST-1). Our findings provide new etiological insights into the pathogenesis of Raine syndrome.

## Results

### FAM20C can regulate GAG abundance

To examine the functional relevance of FAM20C in GAG biosynthesis, we first investigated whether overexpression or knockdown of FAM20C affected GAG production in HeLa cells. GAG fractions isolated from the respective stable clones were treated with bacterial GAG-degrading enzymes, i.e., chondroitinase ABC (ChABC) or a mixture of heparinase and heparitinase. The resulting disaccharides were analyzed using high-performance liquid chromatography (HPLC). As in the case of FAM20B stable clones[23], FAM20C overexpression resulted in significant increases in the total amounts of CS and HS disaccharides, whereas its knockdown had an inverse effect (Fig. 1b, c and Supplementary Table 1). Such changes in GAG content implied biosynthetic alterations in the numbers of GAG chains or the lengths of the individual GAG chains attached to the core proteins[20]. FAM20B upregulates GAG abundance by increasing the number of tetrasaccharide linkers[23]. Gel filtration analysis revealed that FAM20C overexpression increased the length of the CS chains but did not affect the length of the HS chains (Fig. 1d, e). Therefore, FAM20C likely contributes to the elongation of the CS chains. In contrast, the increase in HS abundance due to FAM20C overexpression may have resulted from the increased number of tetrasaccharide linkers. In addition, we found a prominent structural difference in the CS chains, but not in the HS chains, of the stable FAM20C-related HeLa cell lines. Specifically, the elevated expression of FAM20C is associated with an increased proportion of 4-sulfated disaccharide A [GlcA-GalNAc(4-O-sulfate)] units, accompanied by a decreased proportion of 6-sulfated disaccharide C [GlcA-GalNAc(6-O-sulfate)] units, resulting in a relative increase in 4-sulfation/6-sulfation (4S/6S ratio) in the CS chains (Fig. 1f

and Supplementary Table 1). Notably, an opposite trend (i.e., a relative decrease in the 4S/6S ratio) was observed during FAM20B overexpression[23]. Despite significant changes in CS production, the expression profiles of typical CS biosynthetic enzymes were essentially invariable in FAM20C-related stable clones (Supplementary Fig. 1). These results clearly indicate that FAM20C can play additional roles in controlling GAG production levels, possibly because of its distinct functions that overlap with or are independent of those of FAM20B.

To evaluate the overlapping function of FAM20C with that of FAM20B as XYLK, a soluble form of recombinant FAM20C was generated by replacing the 42 N-terminal amino acids of FAM20C with a cleavable insulin signal sequence and a protein A IgG-binding domain and was expressed in COS-1 cells. The FAM20C protein secreted in the medium was adsorbed onto IgG-sepharose beads to eliminate endogenous kinases. Unlike the reported substrate preferences[29], the FAM20C-bound beads showed marked kinase activity toward α-thrombomodulin (α-TM), a native core protein containing linkage tetrasaccharide, and the chemically synthetic acceptor substrates Galβ1-3 Galβ1-4Xylβ1-O-Ser and GlcAβ1–3 Galβ1–3 Galβ1–4Xylβ1-O-SerGlyTrpProAspGly (Supplementary Table 2). The phosphorylation levels and enzymatic characteristics of the secreted FAM20C protein were comparable to those observed in the recombinant FAM20B reactions (Supplementary Fig. 2 and Supplementary Table 2). In contrast, no kinase activity was detected in the reaction products of putative kinase-dead mutants of either FAM20B (D309G) or FAM20C (D478G)[12,13,30] (Supplementary Table 2), substantiating the notion that the respective enzyme-bound beads were highly purified preparations without any other endogenous kinases. These data indicate that FAM20C can act as an XYLK, at least in our in vitro assay conditions.

Phosphorylation of the Xyl residue appears to affect biosynthetic maturation of the linkage tetrasaccharide, regulating the sequential stepwise addition of individual monosaccharides by transfer by the corresponding glycosyltransferases[26,31,32] (Fig. 1a). When using three inter-α-trypsin inhibitors (ITI) preparations that contained sequential intermediates for the linkage saccharides as XYLK substrates (Supplementary Table 3), FAM20C phosphorylated Galβ1–3 Galβ1–4Xylβ1-O-ITI more effectively than Galβ1-4Xylβ1-O-ITI, whereas FAM20B comparably utilized both substrates. For both the FAM20 proteins, Xylβ1-O-ITI was a less effective substrate, supporting a possible inhibitory role of phosphorylated Xyl in the first Gal transfer[26]. Importantly, in metabolically [32P]-labeled stable HeLa cell lines overexpressing either FAM20B or FAM20C, increased levels of two [32P]-labeled linkage saccharide intermediates were detected as fluorophore 2-aminobenzamide (2AB) conjugates, Galβ1-4Xyl(2-O-phosphate)-2AB and Galβ1-3 Galβ1-4Xyl(2-O-phosphate)-2AB: their respective predominant patterns were highly supportive of the in vitro substrate preferences of both FAM20 proteins (Supplementary Fig. 3 and Supplementary Table 3). In contrast, the levels of [32P]-labeled intermediates were clearly decreased by FAM20B or FAM20C knockdown, whereas that of Galβ1-4Xyl(2-O-phosphate)-2AB was unaffected by FAM20C knockdown (Supplementary Fig. 3). These findings indicate that Xyl phosphorylation-dependent GAG biosynthesis is finely controlled by FAM20B and FAM20C.

### XYLK activity-dependent regulation of GAG abundance is not correlated with Raine syndrome

Given that the enzymatic function of FAM20C as an XYLK contributes to Raine syndrome etiology, we expected that mutations in FAM20C would affect its XYLK activity, the phosphorylation status of the Xyl residue in the linkage region, and subsequent GAG production levels. Therefore, we selected four missense mutations found in patients with lethal Raine syndrome: G379R, G379E, L388R, and R549W, located within the kinase domain of FAM20C[2,12,13]. Surprisingly, all mutant FAM20C proteins retained XYLK activity toward the ITI preparations at levels comparable to those detected in wild-type FAM20B and

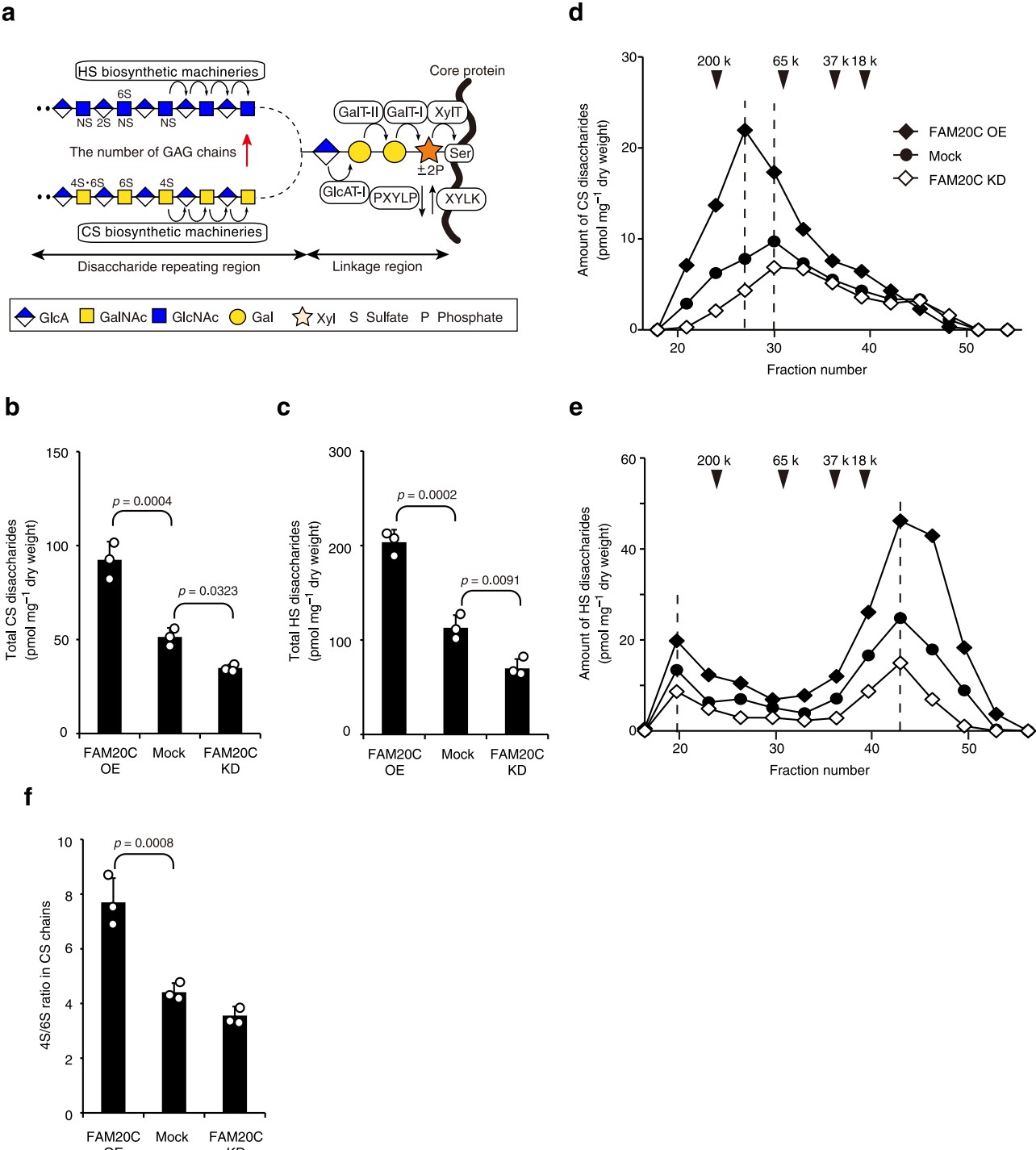

**Fig. 1 | FAM20C can regulate GAG abundance. a** Biosynthetic pathways for CS and HS GAG chains. Multiple glycosyltransferases and distinct kinase/phosphatase(s) contribute to the synthesis of the common tetrasaccharide linkage region and repeating disaccharide region characteristic of CS and HS chains. Transient phosphorylation of the Xyl residue by XYLKs, including FAM20B, enhances linkage tetrasaccharide synthesis. Before subsequent GAG assembly by distinct biosynthetic machinery for CS and HS chains, the Xyl residue is dephosphorylated by a 2-phosphoxylose phosphatase (PXYLP). This XYLK-enhanced GAG biosynthesis determines the number of CS and HS chains attached to core proteins. XylT xylosyltransferase, GalT-I β1,4-galactosyltransferase-I, GalT-II β1,3-galactosyltransferase-II, GlcAT-I β1,3-glucuronyltransferase-I. **b, c** Constitutive overexpression (OE) of FAM20C in HeLa cells elevated the levels of (**b**) CS and (**c**) HS chains, whereas its knockdown (KD) abrogated both GAG levels (**b, c**) compared with the mock controls (*n* = 3 independent experiments, Dunnett's multiple comparison test, two-sided). **d, e** Gel filtration elution profiles of (**d**) CS and (**e**) HS chains from FAM20C-related stable HeLa cell lines. Numbered arrowheads (200 k, 65 k, 37 k, and 18 k) indicate the elution position of 200-, 65-, 37-, and 18-kDa saccharides derived from size-defined commercial dextran, respectively. Results represent the average of two series of independent experiments. **f** The 4S/6S ratio of CS chains from FAM20C-related stable HeLa cell lines (*n* = 3 independent experiments, Dunnett's multiple comparison test, two-sided). Data in **b, c**, and **f** are represented as the mean ± s.d. Source data are provided as a Source Data file.

FAM20C (Supplementary Fig. 4a). When mutant FAM20C proteins were separately overexpressed in HeLa cells, [$^{32}$P]-labeled linkage saccharide intermediates were uniformly detected in each clone, although the fraction of Galβ1-4Xyl(2-O-phosphate)-2AB was always higher than that of Galβ1−3 Galβ1−4Xyl(2-O-phosphate)-2AB (Supplementary Fig. 4b). In addition, overexpression of any mutant protein resulted in higher GAG content in HeLa cells than in mock controls (Supplementary Table 1). We conclude that XYLK activity-dependent GAG biosynthesis is essentially unaffected by Raine syndrome mutations in FAM20C. Nonetheless, gel filtration analysis indicated that the increased CS content due to mutant FAM20C overexpression was not ascribable to enhanced elongation of CS chains (Supplementary Fig. 4c, d), as observed in wild-type FAM20C overexpressing cells. Rather, the substrate preferences of FAM20C mutants and their possible regulatory modes for CS production tended to resemble those of FAM20B[23] (Supplementary Fig. 4).

### FAM20C is involved in the augmentation of 4-sulfation in CS chains

The elongation of CS chains is known to be coupled with their 4-sulfation[33]; thus, we expected that FAM20C-mediated upregulation of CS chains would be controlled by its functional association with 4-sulfation rather than by its own XYLK activity. Supporting this, FAM20C overexpression led to a marked increase in the 4S/6S ratio in CS chains (Fig. 1f and Supplementary Table 1). Such sulfation profiles can change during physiological processes[34,35], and we have previously demonstrated that the 4S/6S ratio is essential for terminating the critical period for ocular dominance plasticity[36]. Given the biological importance of the 4S/6S ratio, we next focused on the functional relevance of FAM20C in regulating the 4S/6S ratio of the CS chains.

During CS biosynthesis, GalNAc residues in the repeating disaccharide units are sulfated at the C4 and C6 positions by chondroitin 4-O-sulfotransferases (C4ST-1 and C4ST-2) and chondroitin 6-O-sulfotransferase-1 (C6ST-1), resulting in the formation of A and C units, respectively[20]. As subsequent sulfation of the monosulfated disaccharide units also occurs by additional enzymes responsible for the generation of the disulfated disaccharide units E [GlcA-GalNAc(4,6-O-disulfate)] and D [GlcA(2-O-sulfate)-GalNAc(6-O-sulfate)], sulfation of the chondroitin backbone can be classified into "4-sulfation" and "6-sulfation" pathways[20] (Fig. 2a). To find evidence for our hypothesis, we utilized murine L cells and a mutant cell lineage, sog9, which is deficient in *C4st1* expression and therefore produces CS chains with a low 4S/6S ratio[37] (Supplementary Table 4). As expected, wild-type FAM20C overexpression increased the lengths of CS chains in the L cells, but not in sog9 cells (Fig. 2b). No such changes were observed in either cell type overexpressing FAM20B (Fig. 2b), suggesting a selective requirement for C4ST-1 during FAM20C-mediated chain elongation. Indeed, the pulldown assay revealed that FAM20C, but not FAM20B, physically interacted with C4ST-1 when co-expressed in COS-1 cells (Fig. 2c). An in vitro kinase assay showed that FAM20C could phosphorylate recombinant C4ST-1 (Supplementary Fig. 5a), which has a consensus S-X-E/pS motif ($^{323}$SAE) for protein phosphorylation by FAM20C[12,15], indicating that C4ST-1 can serve as a substrate and therefore may physically interact with wild-type FAM20C. Co-expressed FAM20C and C4ST-1 also exhibited relatively higher sulfotransferase activity toward chondroitin (a chemically desulfated form of CS polysaccharide) than either C4ST-1 alone or with FAM20B; however, this difference was not statistically significant ($p = 0.0595$) (Fig. 2d). These results indicate that FAM20C can interact with and augment the sulfotransferase activity of C4ST-1.

Next, we tested whether lethal Raine syndrome mutations in FAM20C affected C4ST-1-mediated CS biosynthesis. Notably, none of the four mutant proteins interacted with C4ST-1, despite their expression levels being comparable to their wild-type FAM20C counterparts (Fig. 2c). Supporting this observation, none of the mutant

proteins could enhance the sulfotransferase activity of C4ST-1 when co-expressed in COS-1 cells (Fig. 2d). Furthermore, no marked increase of the 4S/6S ratio was observed in stable HeLa cells overexpressing any of these mutants (Fig. 2e). Taken together, these results suggest that the 4S/6S ratio in CS chains may be perturbed in the lethal forms of Raine syndrome.

### FAM20C mutants downregulate the 4S/6S ratio in CS chains and enhance biomineralization in human osteosarcoma cells

To clarify the etiology of Raine syndrome, we examined the impact of FAM20C mutations on human osteogenic cells. In the present study, we found that the human osteosarcoma Saos-2 cell line[38,39] is deficient in FAM20C (Fig. 3a, b). Taking advantage of this characteristic, we established Saos-2 cells that stably overexpressed individual FAM20 proteins. Notably, though wild-type FAM20C overexpression did not augment the 4S/6S ratio in these cells compared to that in mock control cells, the overexpression likely contributed to the maintenance of a high 4S/6S ratio (Fig. 3c). In contrast, overexpression of wild-type FAM20B or any of the FAM20C mutants downregulated the 4S/6S ratio (Fig. 3c). In pulldown assays using Saos-2 cell lysates, we confirmed a mutually exclusive interaction between wild-type FAM20C and endogenous C4ST-1 (Supplementary Fig. 5b). Most importantly, biomineralization in Saos-2 cells was significantly enhanced by overexpression of wild-type FAM20B or any of the FAM20C mutants, but not by overexpression of wild-type FAM20C (Fig. 3d, e). The apparent phenotypic similarities between wild-type FAM20B and FAM20C mutants in Saos-2 cell culture suggest that FAM20C mutants could behave as FAM20B-like proteins instead of losing distinct FAM20C properties. These results prove that lethal Raine syndrome mutations in FAM20C could be the etiological factor in osteosclerotic bone dysplasia, by causing decreased 4S/6S ratio in the CS chains.

### Upregulation of chondroitin 6-sulfation promotes osteoblast differentiation

The engineering and characterization of osteoblasts with decreased 4S/6S ratio in CS chains could help understand the etiological role of this decreased ratio in Raine syndrome. One potential approach, the functional knockdown of C4ST-1, is well-known but often leads to a prominent decrease in CS abundance rather than a decrease in the 4S/6S ratio[33,37,40–42]. Furthermore, considering that the proportion of 6-sulfated CS chains of mouse origin is relatively lower than that of chains of human origin (Fig. 1f, 3c, and Supplementary Tables 1 and 4), we chose an alternative way to overexpress C6ST-1 in mouse MC3T3-E1 osteoblasts, an excellent model for studying membranous ossification[43].

The 4S/6S ratio in C6ST-1 overexpressed MC3T3-E1 cells decreased without any significant change in the CS content compared to that in the parental or mock control cells (Fig. 4a and Supplementary Table 5). When induced to differentiate, C6ST-1 overexpressing MC3T3-E1 cells exhibited a marked increase in the expression of *Akp2*, a typical osteogenic marker gene for alkaline phosphatase (ALP), and mineral deposition (Fig. 4b, c). Osteoblast differentiation of MC3T3-E1 cells is reportedly initiated by reciprocal cell–cell contact mediated by N-cadherin and cadherin-11[44,45]. We previously demonstrated that CS-E, a CS subtype rich in E units, can bind to both these cadherins and stimulate cadherin-mediated intracellular signaling pathways integral to osteoblast differentiation[46]. Similarly, CS-C, a typical 6-sulfated CS preparation, is also bound to N-cadherin and cadherin-11 with notable affinities but only in the presence of calcium ions (Supplementary Fig. 6). Adhesion of MC3T3-E1 cells to culture plates precoated with N-cadherin or cadherin-11 was enhanced by C6ST-1 overexpression; the effect of C6ST-1 overexpression was attenuated by pretreatment of the cells with ChABC and was largely eliminated in the presence of EDTA (Fig. 4d). In addition, exogenous application of CS-C stimulated the cadherin-mediated onset of osteoblast differentiation of parental MC3T3-E1 cells (Fig. 4e), as assessed by reduced ERK1/2 phosphorylation[47],

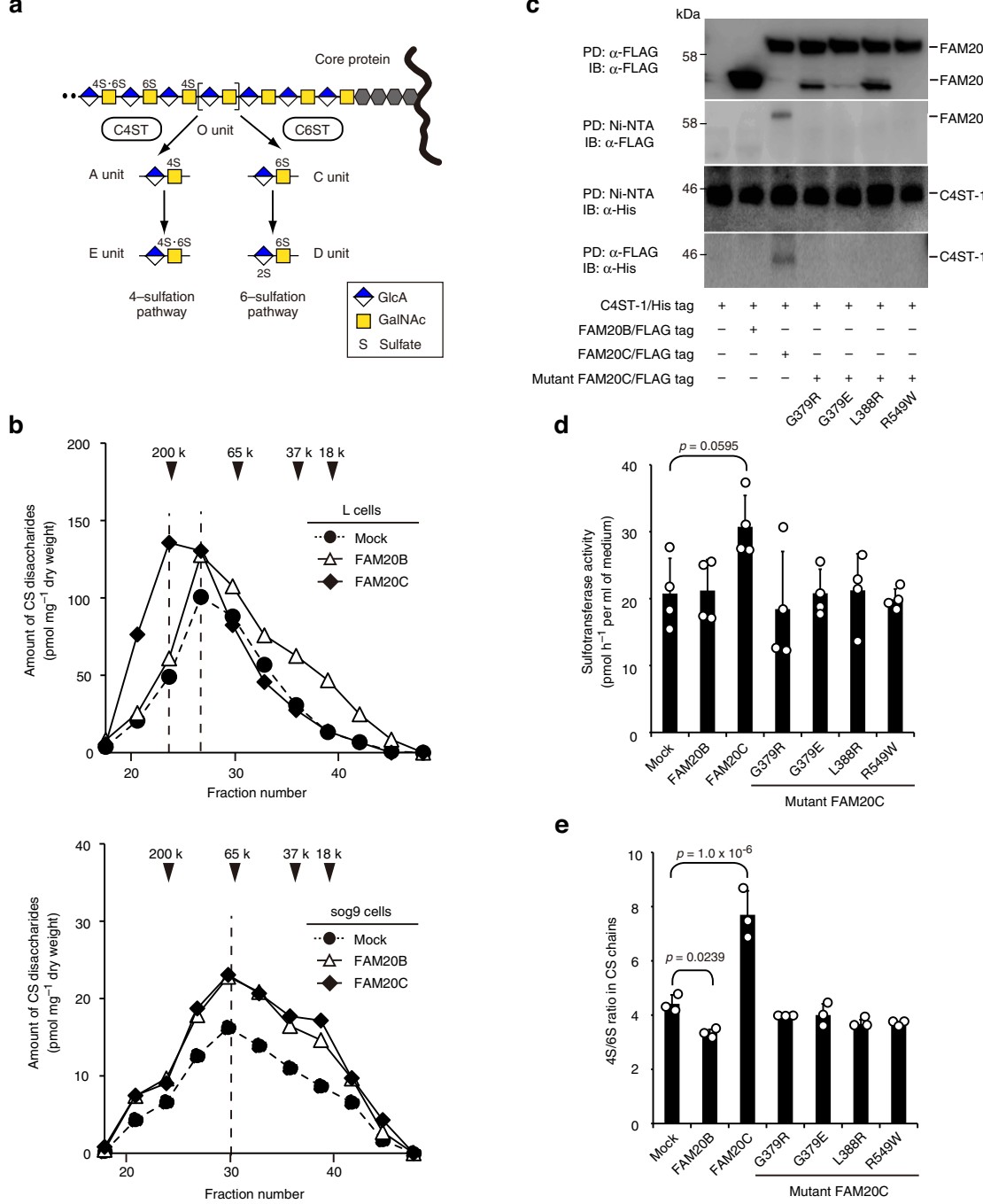

**Fig. 2 | FAM20C-mediated control of CS biosynthesis is exerted by its physical interaction with C4ST-1. a** Schematic diagram of sulfation pathways for the chondroitin backbone. Characteristic CS disaccharide units, including A, C, D, and E, are sequentially formed under the control of CS-specific sulfotransferases such as C4ST-1, C4ST-2, and C6ST-1. **b** Gel filtration elution profiles of CS chains from mouse fibroblastic L or sog9 cells overexpressing wild-type FAM20B or FAM20C (Data are derived from the single experiment). Numbered arrowheads (200 k, 65 k, 37 k, and 18 k) indicate the elution position of 200-, 65-, 37-, and 18-kDa saccharides derived from size-defined commercial dextran, respectively. **c** Pulldown (PD) and immunoblotting (IB) analysis of culture medium from COS-1 cells co-expressing

soluble forms of His-tagged C4ST-1 and FLAG-tagged FAM20 proteins. Data are obtained from three independent experiments and representative images are shown. **d** Sulfotransferase activities of C4ST-1 alone (Mock) and that from cells co-transfected with the individual FAM20 proteins ($n = 3$ independent experiments, Dunnett's multiple comparison test, two-sided). **e** The 4S/6S ratio of CS chains from HeLa cell lines overexpressing wild-type or mutant FAM20C proteins ($n = 4$ independent experiments, Dunnett's multiple comparison test, two-sided). Data in **d** and **e** are represented as the mean ± s.d. Source data are provided as a Source Data file.

increased Smad3 and Smad1/5/8 phosphorylation[48,49], and increased *Akp2* expression. These results suggest that upregulated chondroitin 6-sulfation can accelerate osteoblast differentiation, implying an etiologic role for the decreased 4S/6S ratio of CS chains in osteosclerotic bone dysplasia.

## *C6ST1* transgenic mice show a high BMD

Next, we attempted to verify whether C6ST-1 overexpression contributes to the manifestation of osteosclerotic phenotypes in vivo. To this end, we characterized the long bones from *C6ST1* transgenic mice[36]. At 16 weeks of age, the long bones of male *C6ST1* transgenic

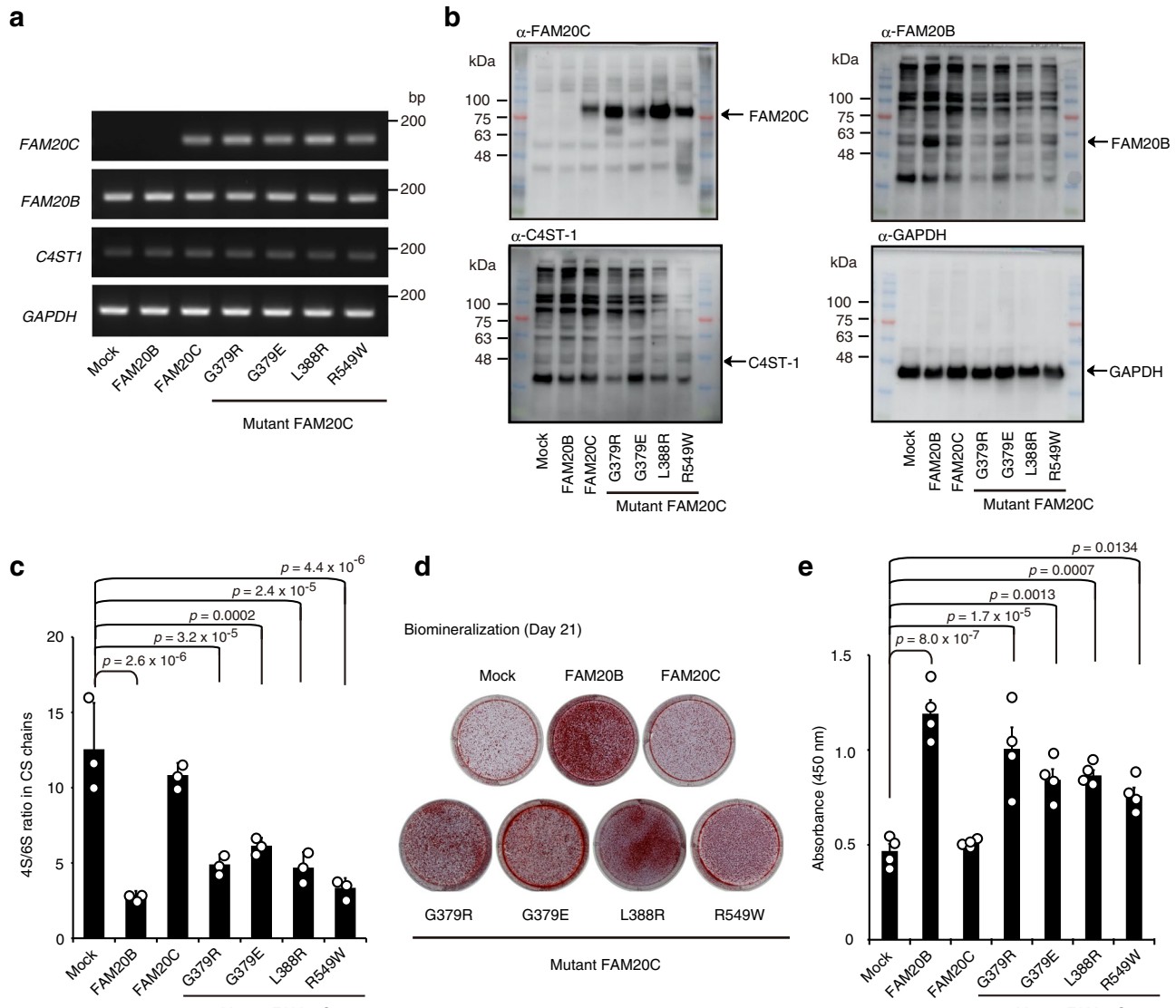

**Fig. 3 | FAM20C mutants enhance biomineralization in human osteosarcoma cells. a, b** Expression of FAM20C, FAM20B, and C4ST-1 in stable Saos-2 cell lines overexpressing individual FAM20 proteins at the **a** transcriptional and **b** translational levels. GAPDH was used as an internal standard in (**a**), and as a loading control in (**b**). Data are obtained from three independent experiments and representative images are shown. **c** The 4S/6S ratio of CS chains from the stable Saos-2 cell lines in (**a**) and (**b**) ($n = 3$ independent experiments, Dunnett's multiple comparison test, two-sided). **d, e** Biomineralization level of the 21-day cultures of the stable Saos-2 cell lines in (**a**) and (**b**) was assessed by Alizarin red staining (**d**), followed by colorimetric quantification of the dye extracts derived from the stained cultures (**e**, $n = 4$ independent experiments, Dunnett's multiple comparison test, two-sided). Data in **c** and **e** are represented as the mean ± s.d. Source data are provided as a Source Data file.

mice were significantly longer than those of their wild-type littermates (Fig. 5a). Peripheral quantitative computed tomography (CT) and dual-energy X-ray absorptiometry revealed that *C6ST1* transgenic tibias had significantly higher BMD than the wild-type control tibias (Supplementary Fig. 7a–d). Comparison of trabecular bone parameters and cortical bone thickness of the respective tibias obtained by μ-CT analysis also supported the high bone mass phenotypes of *C6ST1* transgenic mice (Fig. 5b–i). The higher BMD in *C6ST1* transgenic mice was accompanied by a significant increase in bone mechanical strength (Supplementary Fig. 7e).

To further investigate whether the high bone mass phenotypes of *C6ST1* transgenic mice were due to the acceleration of bone anabolic processes, the osteoblastic potential of bone marrow-derived stromal cells (BMSCs) from the *C6ST1* transgenic mice was analyzed. The *C6ST1* transgenic BMSCs produced CS chains with significantly reduced 4S/6S ratios (Fig. 5j and Supplementary Table 6). Compared with wild-type controls, osteoblastogenesis was significantly enhanced in *C6ST1*

transgenic BMSCs, as assessed by the levels of ALP expression and mineral deposition (Fig. 5k–m). Depletion of CS chains by ChABC treatment severely suppressed osteoblast differentiation in both the transgenic and control mice BMSCs (Fig. 5k–m). These results indicate that an unusual decrease in the 4S/6S ratio of CS chains might lead to the formation of osteosclerotic phenotypes that are characteristic of lethal Raine syndrome.

## Discussion

Here, we demonstrated novel functions of FAM20C in GAG biosynthesis, which can improve our knowledge of the pathogenesis of osteosclerotic bone dysplasia. This form of dysplasia in humans is referred to as the Raine syndrome. In contrast to the data from previous reports[12–15,29], data from the present study emphasize that FAM20C not only acts as a Golgi casein kinase but also as an XYLK, similar to FAM20B, which controls GAG abundance by providing the linkage tetrasaccharide primer for subsequent GAG assembly (Fig. 1a).

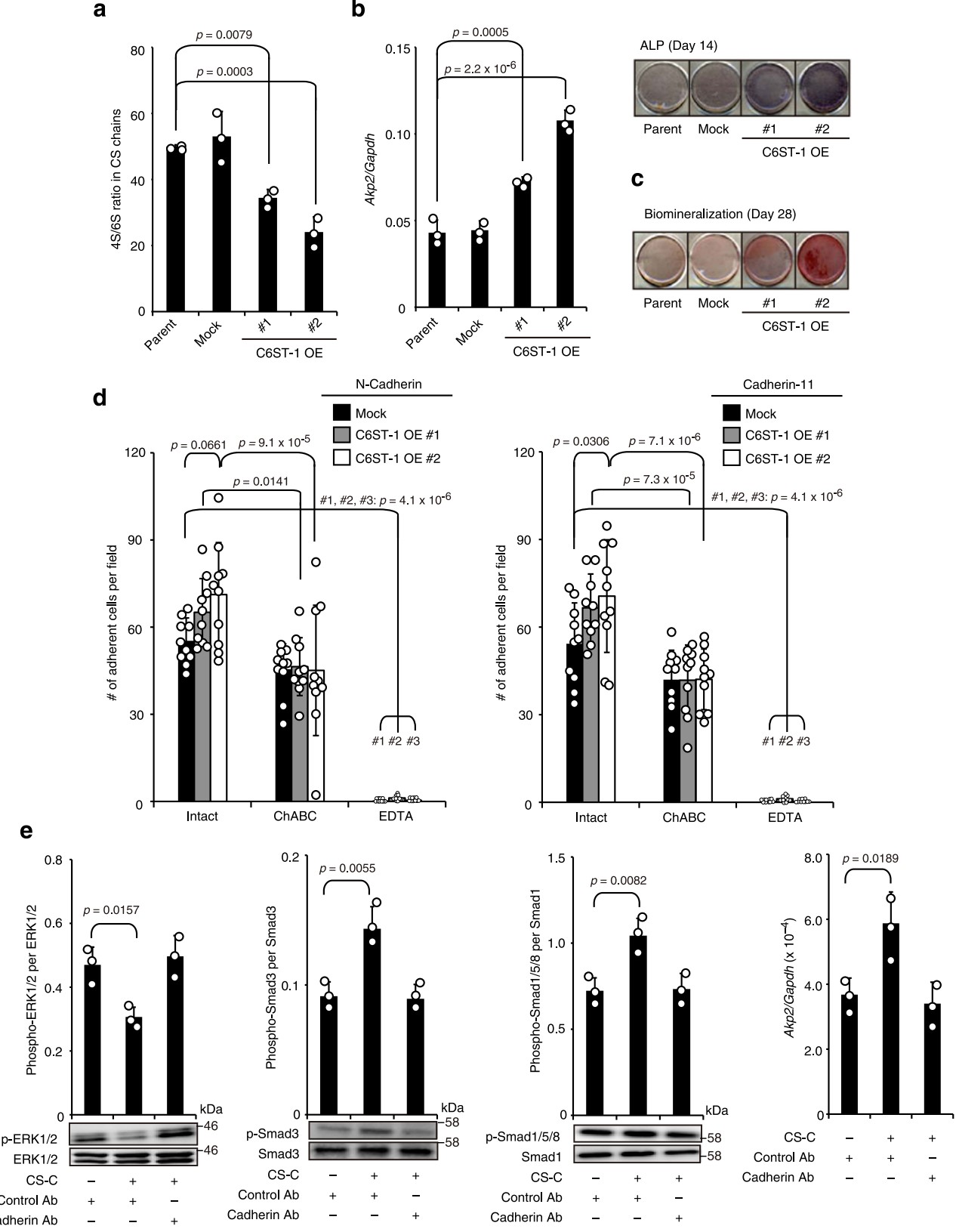

The apparent inconsistency from previous reports may be attributed to the structural differences in substrates used for in vitro assays, because the aglycone moieties of substrates are known to affect GAG biosynthetic efficiency[50]. We also revealed the non-enzymatic role of FAM20C in fine-tuning the sulfation status of the CS chains by forming a complex with C4ST-1. FAM20C augmented C4ST-1 activity and produced CS chains with a high 4S/6S ratio (Fig. 6a). The kinase-independent actions of FAM20C were most interesting to be explored a potential Raine syndrome etiology because the decreased 4S/6S ratio caused by lethal Raine syndrome mutations in FAM20C was well associated with osteosclerotic status, even in mouse models (Fig. 6b). Importantly, the functional characteristics of the mutant FAM20C proteins observed in this study were similar to those of FAM20B protein[23] (Figs. 2, 3 and Supplementary Fig. 4). FAM20B has

**Fig. 4 | C6ST-1 overexpression potentiates mouse osteoblast differentiation.**
**a** The 4S/6S ratio of CS chains from parental, mock-transfected (Mock), or C6ST-1 overexpressing (C6ST-1 OE #1 and #2) MC3T3-E1 cells ($n = 3$ independent experiments, Dunnett's multiple comparison test, two-sided). **b** ALP staining and mRNA expression for ALP (*Akp2*) in 14-day cultures of parental, mock, or C6ST-1 OE cells (#1 and #2) ($n = 3$ independent experiments, Dunnett's multiple comparison test, two-sided). **c** Mineralized nodule formation in 21-day cultures of parental, mock, or C6ST-1 OE cells (#1 and #2) was assessed by Alizarin red staining. Data are obtained from three independent experiments and representative images are shown. (**d**) Adhesion of intact, ChABC-pretreated, or EDTA-treated MC3T3-E1 cells (parental or C6ST-1 OE #1 and #2) to plates precoated with recombinant N-cadherin or cadherin-11 ($n = 10$ fields from 3 independent experiments, for each condition, Tukey–Kramer multiple comparison method). **e** CS-C modulated the cadherin-mediated intracellular signaling pathways, including ERK1/2, Smad3, and Smad1/5/8, and upregulated *Akp2* expression in semi-confluent, parental MC3T3-E1 cultures in the presence of an isotype control antibody (control Ab). Pretreatment with neutralizing antibodies for N-cadherin and cadherin-11 (cadherin Ab) abolished the CS-C effects ($n = 3$ independent experiments, Dunnett's multiple comparison test, two-sided). Data in **a**, **b**, **d**, and **e** are represented as the mean ± s.d. Source data are provided as a Source Data file.

been reported as a psoriasis-risk gene[51]. The impact of the CS sulfation profile on the pathogenesis of psoriasis has been demonstrated and provides strong support for the kinase-independent role of FAM20B in the augmentation of chondroitin 6-sulfation, resulting in a decreased 4S/6S ratio[52]. Therefore, the imbalanced 4S/6S ratio caused by FAM20C mutations might be attributed not only to their lack of ability to interact with C4ST-1 but also to their acquisition of FAM20B-like properties to increase chondroitin 6-sulfation (Fig. 6b).

The 4S/6S ratio of CS chains is generally much higher in mice than in humans, most likely due to the relatively low expression of chondroitin 6-sulfation in mice (Figs. 1f, 3c, 4a, 5j, Supplementary Tables 1, 4–6). Such a difference between species is believed to result in phenotypic variation between humans and mice based on C6ST-1 deficiency. Indeed, loss-of-function mutations in human *C6ST1* are associated with skeletal disorders, including spondyloepiphysial dysplasia and lumbar disc degeneration[53,54], whereas *C6ST1* knockout mice exhibit no apparent abnormalities[55]. These differences are not surprising, given that a certain threshold 4S/6S ratio is required for the participation of CS in skeletal development. Although currently, there is no direct evidence for this view, this idea may also aid in understanding why FAM20C deficiency in mice models cannot recapitulate human osteosclerotic phenotypes; that is, the threshold 4S/6S ratio level required for the manifestation of chondroitin 6-sulfation-dependent skeletal phenotypes might not be achievable by FAM20C mutations but only by C6ST-1 overexpression in mice.

Previously, we demonstrated that CS-E promoted in vitro osteoblastogenesis[46] and participated in estrogen-induced osteoanabolism in vivo[56]. Although the E units possess a 6-sulfated structure, they are synthesized via the "4-sulfation" pathway[20] (Fig. 2a). As FAM20C can positively regulate the "4-sulfation" pathway, it may also contribute to CS-E-mediated normal bone formation. In contrast to our findings, Hao et al. reported that FAM20C/DMP4 overexpression promoted the mineralization of MC3T3-E1 cell cultures, whereas its knockdown suppresses their osteoblast differentiation[11]. These apparently contradictory results might be attributed to the supportive function of FAM20C in CS-E-mediated osteoanabolism under conditions of low levels of chondroitin 6-sulfation. Therefore, the physiological or pathological functions of FAM20C in the bone formation may be substantially affected by tissue- or species-specific basal levels of chondroitin 6-sulfation.

Cui et al. previously described the non-enzymatic function of FAM20A as an allosteric activator of the protein kinase activity of FAM20C[57]. This strengthens the theory of the multifunctional nature of FAM20 proteins, including their enzymatic and non-enzymatic actions in GAG biosynthesis. Therefore, further studies on FAM20 family molecules are needed for a comprehensive understanding of the biosynthetic machinery that fine-tunes GAG chains and for a complete picture of Raine syndrome etiologies.

## Methods
### Materials
[γ-³²P]ATP (3,000 Ci mmol⁻¹), [γ-³²P]NaH₂PO₄ (285.2 mCi mmol⁻¹), and [³⁵S]PAPS (adenosine 3′-phosphate 5′-phosphosulfate) (1.69 mCi mmol⁻¹) were purchased from PerkinElmer (Waltham, MA, USA). Unlabeled ATP was purchased from Sigma-Aldrich (St. Louis, MO, USA). *Proteus vulgaris* chondroitinase ABC (EC 4.2.2.4), *Flavobacterium heparinum* heparinase (EC 4.2.2.7), heparitinase (EC 4.2.2.8), shark cartilage CS-C, and chondroitin (a chemically desulfated derivative of CS-C) were purchased from Seikagaku Corp. (Tokyo, Japan). α-TM with a truncated linkage region tetrasaccharide, GlcAβ1–3 Galβ1–3 Galβ1–4Xyl, was purified and structurally characterized[58]. Galβ1–3 Galβ1–4Xyl(2-*O*-phosphate)β1-*O*-Ser[59] and GlcAβ1–3 Galβ1–3 Galβ1–4Xylβ1-*O*-SerGly TrpProAspGly[60] were chemically synthesized. Galβ1–3 Galβ1–4Xylβ1-*O*-Ser was prepared by digestion of Galβ1–3 Galβ1–4Xyl(2-*O*-phosphate)β1-*O*-Ser with alkaline phosphatase[23]. Mouse L-fibroblasts and their mutant derivatives, sog9 cells, were kindly provided by Dr. Frank Tufaro (Allera Health Products, St. Petersburg, FL, USA). HeLa cells (ATCC®, CCL-2™) and COS-1 cells (ATCC®, CRL-1650™) were obtained from American Type Culture Collection (ATCC). The human osteosarcoma cell line, Saos-2 (RCB0428), and mouse osteoblastic cell line, MC3T3-E1 (RCB1126), were provided by the RIKEN BRC through the National Bio-Resource Project of the MEXT/AMED, Japan. Antibodies used in this study are listed in Supplementary Table 7.

### Mice
*C6ST1* transgenic mice[36] (C57BL/6 genetic background), were kept under specific pathogen-free conditions in an environmentally controlled (23 ± 1 °C with 50 ± 10% humidity), bio-clean room at the Institute of Laboratory Animals, Kobe Pharmaceutical University, and maintained in standard rodent food and a 12-h light–dark cycle. All experiments were conducted according to the institutional ethics guidelines for animal experiments and the safety guidelines for gene manipulation experiments of Kobe Pharmaceutical University. All animal procedures were approved by the Kobe Pharmaceutical University Committee on Animal Research and Ethics. After weaning, *C6ST1* transgenic mice and their wild-type littermates were housed separately. Male mice aged at 8 and 16 weeks were used for bone analyses and BMSC preparation, respectively. For these purposes, CO₂ euthanasia was performed.

### Generation of stably transfected HeLa and mouse fibroblast cell lines
Full-length human *FAM20C* cDNA was polymerase chain reaction (PCR)-amplified using a cDNA clone (IMAGE, 4942737) as the template and specific primers, both containing a *Bgl*II site. The PCR fragment was subcloned into the *Bgl*II site of a pCMV expression vector (Invitrogen, Life Technologies, Carlsbad, CA, USA). The fidelity of the plasmid construct (pCMV-FAM20C) was confirmed by DNA sequencing. Expression plasmids encoding the mutant FAM20C proteins were constructed using a two-stage PCR mutagenesis method. Two overlapping gene fragments for the corresponding mutant FAM20C proteins were separately amplified by PCR using pCMV-FAM20C as a template and two distinct primer sets one carrying the above-mentioned sense primer with a *Bgl*II site and the antisense internal mutagenic (IM) primer, and the other carrying the sense IM primer (complementary to the antisense IM primer) and the above-mentioned

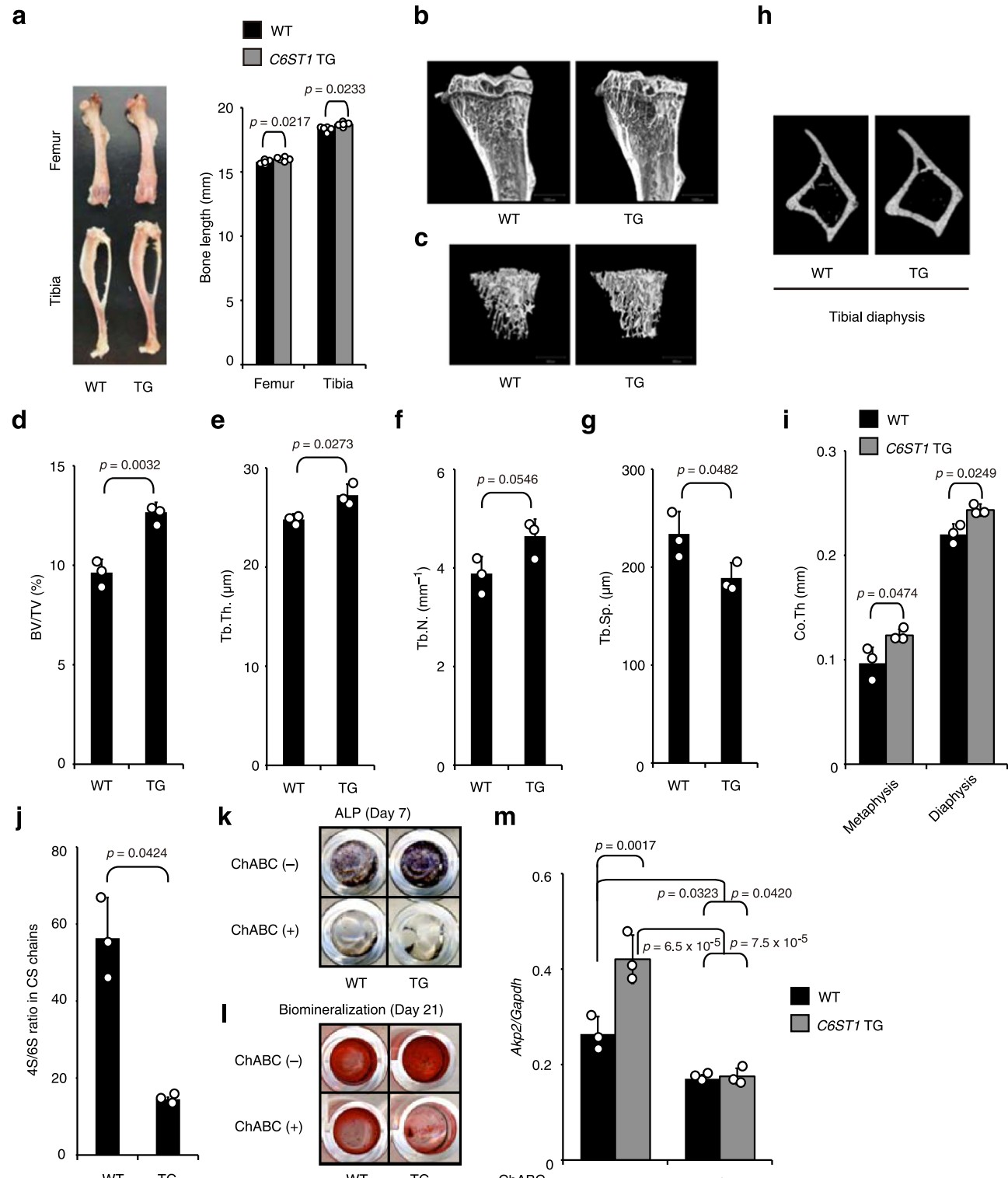

**Fig. 5 | High bone mass phenotypes in *C6ST1* transgenic mice. a** The length of long bones (femurs and tibias) from 16-week-old wild-type (WT) or *C6ST1* transgenic (TG) male mice (*n* = 5 bones from total, each from different litters, unpaired Student's *t*-test, two-sided). **b–i** Micro-CT analysis of tibias from 16-week-old WT or *C6ST1* TG male mice. **b** Medial, longitudinal section through a µCT-generated three-dimensional reconstruction of a tibia. **c** Micro-CT reconstruction of the trabecular region of tibial metaphysis. Micro-CT-based measurements of **d** trabecular bone volume to total volume (BV/TV), **e** trabecular thickness (Tb. Th.), **f** trabecular number (Tb. N.), and **g** trabecular separation (Tb. Sp.). **d–g** *n* = 3 bones from the total, each from different litters, unpaired Student's *t*-test, two-sided). **h** Micro-CT reconstruction of the cortical region of the tibial diaphysis. **i** Micro-CT-based measurement of cortical thickness (Co. Th.). (*n* = 3 bones from total, each from

different litters, unpaired Student's *t*-test, two-sided). **j–m** Decreased 4S/6S ratio of CS chains promotes osteoblastogenesis. **j** The 4S/6S ratio of CS chains from WT or *C6ST1* TG BMSCs (*n* = 3 cultures from 3 independent mice litters, for each genotype, unpaired Student's *t*-test, two-sided). **k–m** Osteoblastic potential of BMSCs isolated from WT or *C6ST1* TG male mice. BMSCs were maintained in a differentiation medium (DM) for 21 days in the absence (−) or presence (+) of ChABC. ALP staining (**k**) and Alizarin red staining (**l**) were performed on 7-day and 21-day cultures of BMSCs, respectively. Expression of *Akp2* mRNA in 7-day cultures of BMSCs (**m**, *n* = 3 cultures from 3 independent mice litters, for each genotype, Tukey–Kramer multiple comparison method). Data in **a**, **d–g**, **i**, **j**, and **m** are represented as the mean ± s.d. Source data are provided as a Source Data file.

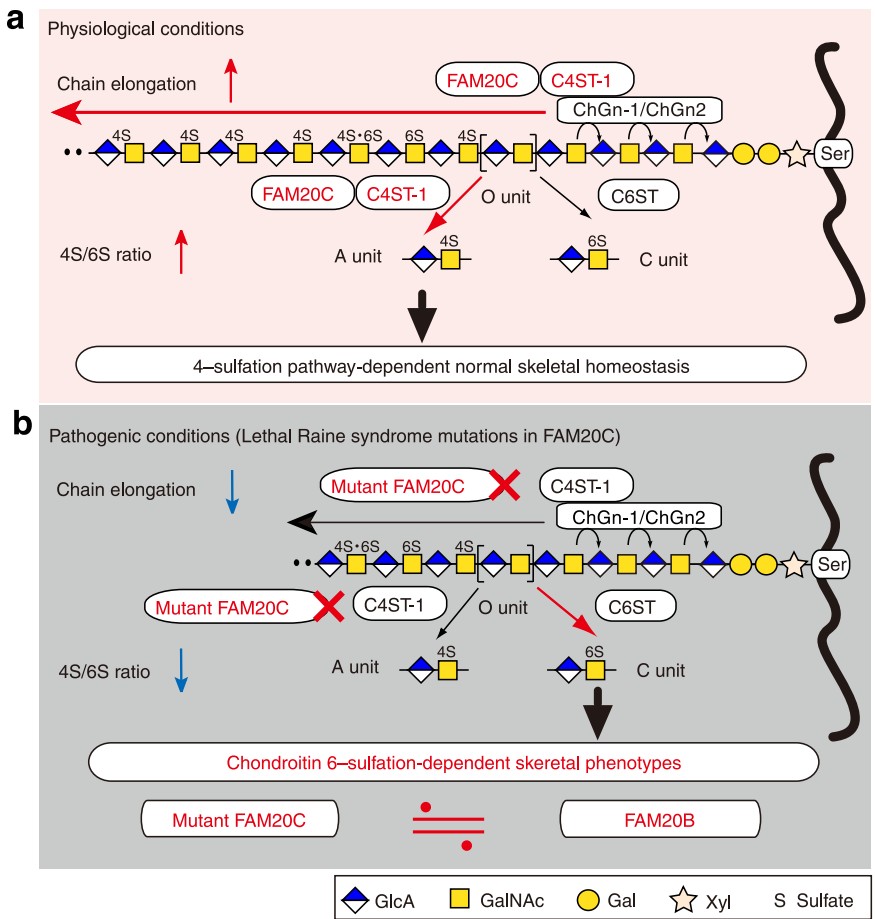

**Fig. 6 | Non-enzymatic functions of FAM20C in the control of CS biosynthesis and its involvement in Raine syndrome etiology.** In physiological states (**a**), FAM20C can physically interact with C4ST-1 and augment the enzymatic activity of C4ST-1, leading to increased length of the individual CS chains via cooperative actions with glycosyltransferases (*N*-acetylgalactosaminyltransferases; ChGns)[20,33] and to an elevated 4S/6S ratio. In contrast **b**, lethal Raine syndrome mutations in FAM20C cause functional uncoupling between FAM20C and C4ST-1, perturbing the C4ST-1-mediated chain elongation process and sulfation balance of CS chains. The consequent decrease in the 4S/6S ratio may be involved in the manifestation of chondroitin 6-sulfation-dependent skeletal phenotypes, which might be recapitulated by a gain-of-function of FAM20B.

antisense primer with a *Bgl*II site. To obtain full-length fragments of the respective mutant *FAM20C* genes, these two PCR products were used as templates for subsequent PCR using the same primer set, which was used for pCMV-FAM20C construction. The final PCR fragment was subcloned into the *Bgl*II site of the pCMV expression vector and the resultant constructs (pCMV-FAM20Cs G379R, G379E, L388R, and R549W) were sequenced. The primer sequences used for the plasmid construction are listed in Supplementary Table 8.

The expression plasmids for FAM20B (pCMV-FAM20B)[23], FAM20C (pCMV-FAM20C), or the respective FAM20C mutants were individually transfected into HeLa cells using FuGENE™ 6 transfection reagent (Promega, Madison, WI, USA), according to the manufacturer's instructions. Clonal cell lines were selected and grown in the presence of 300 μg ml⁻¹ G418. For stable knockdown, validated MISSION shRNA plasmid vectors targeting FAM20B (TRCN0000138872, NM_014864) or FAM20C (TRCN0000128300, NM_020223) (Sigma-Aldrich) were individually transfected into HeLa cells using the FuGENE™ 6. Stable transfectants were selected and propagated in the presence of 0.4 μg ml⁻¹ puromycin, along with stable cell lines derived from mouse fibroblastic L and sog9 cells with the expression plasmids pCMV-FAM20B or pCMV-FAM20C.

## Soluble FAM20 protein expression
The cDNA fragment of the truncated FAM20C sequence lacking the first 42 N-terminal amino acids was PCR-amplified with pCMV-FAM20C

as the template using a primer set designed to have *Bgl*II sites at each primer (Supplementary Table 8). The PCR fragment was then subcloned into the *Bam*HI site of pGIR201proA[61], resulting in the fusion of the insulin signal and the protein A sequences present in the vector (pEF-BOS/IP-FAM20C). Soluble FAM20C mutants were constructed by PCR-based site-directed mutagenesis as described above. Resulting plasmid constructs (pEF-BOS/IP-FAM20Cs, D478G, G379R, G379E, L388R, and R549W) were confirmed for fidelity by sequencing. A kinase-dead mutant of FAM20B (D309G) was constructed using site-directed mutagenesis.

The expression vectors (6.0 μg), including pEF-BOS/IP-FAM20B[23] and pEF-BOS/IP-FAM20Cs, were individually transfected into COS-1 cells on 100-mm plates using FuGENE™ 6, as described above. For the co-transfection experiments, pEF-BOS/IP-C4ST-1[62] and each of the FAM20-related expression plasmids (3.0 μg each) were used. Two days after transfection, 1 ml of culture medium was collected and incubated with 10 μl of IgG-sepharose (Cytiva, Tokyo, Japan) for 1 h at 4 °C. The beads were recovered by centrifugation, washed, resuspended in assay buffer, and tested for XYLK and sulfotransferase activities[23,62].

## ITI preparations
To create linkage saccharide intermediates for XYLK assays, a synthetic peptide GlnGluGluGluGlyySerGlyGlyGlyGlnLys (QEEEGSGGGQK) derived from ITI (Invitrogen) was used as an acceptor for sequential stepwise addition of individual monosaccharide units, single Xyl, and

two successive Gal residues by the corresponding specific glycosyltransferases, xylosyltransferase-I (XylT-I)[63], GalT-I[31], and GalT-II[64], respectively. To obtain recombinant enzymes, expression constructs of glycosyltransferases were prepared. The cDNA fragment of truncated XylT-I, lacking the first 148 N-terminal amino acids, was amplified by PCR using specific primers, each having a *Bgl*II site (Supplementary Table 8). The cDNA fragment of truncated GalT-I, lacking the first 53 N-terminal amino acids, was amplified by PCR using *Bam*HI site-containing specific primers (Supplementary Table 8). The cDNA fragment of the truncated GalT-II sequence, lacking the first 34 N-terminal amino acids, was amplified by PCR using specific primers containing a *Bam*HI site each (Supplementary Table 8). Each PCR fragment was subcloned into the *Bam*HI site of pGIR201proA (pEF-BOS/IP-XylT-I, -GalT-I, and -GalT-II).

In the first step, the synthetic peptide QEEEGSGGGQK (10 nmol) was used as an acceptor in the incubation mixture containing 10 μl of beads bound to the soluble form of XylT-I, 25 mM MES buffer (pH 6.5), 25 mM KF, 5 mM $MgCl_2$, 5 mM $MnCl_2$, and 2 mM UDP-[$^{14}$C]Xyl ($3.0 \times 10^5$ dpm) (PerkinElmer). The reaction was performed at 37 °C for 16 h. The reaction products containing the xylosylated peptide (Xyl-*O*-ITI) were further used as acceptors in an incubation mixture containing 10 μl of beads bound to the soluble form of FAM20B[23], 50 mM Tris/HCl, pH 7.0, 10 mM $MnCl_2$, 10 mM $CaCl_2$, and 10 μM [γ-$^{32}$P]ATP ($1.11 \times 10^5$ dpm), to generate Xyl(2-*O*-phosphate)-*O*-ITI. The first Gal transfer to Xyl(2-*O*-phosphate)-*O*-ITI was conducted in a mixture containing 20 μl beads bound to the soluble form of GalT-I, 100 mM acetate buffer (pH 6.5), 5 mM $MnCl_2$, and 1 mM UDP-Gal. The second Gal transfer to the resultant products was performed in a mixture containing an excess of beads bound to the soluble form of GalT-II, 50 mM MES (pH 6.0), 15 mM $MnCl_2$, 50 mM KCl, and 10 mM UDP-Gal. The products of each reaction were then treated with rAPid alkaline phosphatase (Roche Diagnostics, Mannheim, Germany) to dephosphorylate the 2-*O*-phosphorylated saccharide-ITI preparations. The reactions were stopped by heat treatment and subjected to gel filtration using a Superdex™ Peptide HR 10/30 column (Cytiva) equilibrated with 0.25 M $NH_4HCO_3$/7% 1-propanol. Fractions containing enzyme reaction products were pooled and dehydrated. The isolated reaction products were used as substrates for XYLK reactions.

## XYLK assay

XYLK reactions were conducted using α-TM and chemically synthesized linkage saccharide preparations (Galβ1-3 Galβ1-4Xylβ1-*O*-Ser, Galβ1-3 Galβ1-4Xyl(2-*O*-phosphate)β1-*O*-Ser, and GlcAβ1−3 Galβ1−3 Galβ1−4Xylβ1-*O*-SerGlyTrpProAspGly) or enzymatically synthesized ITI preparations (Xylβ1-*O*-ITI, Galβ1-4Xylβ1-*O*-ITI, and Galβ1−3 Galβ1−4Xylβ1-*O*-ITI) as acceptors and ATP as a phosphate donor. The standard reaction mixture (20 μl) contained 10 μl of the resuspended beads, 50 mM Tris/HCl, pH 7.0, 10 mM $MnCl_2$, 10 mM $CaCl_2$, 10 μM [γ-$^{32}$P]ATP ($1.11 \times 10^5$ dpm), and an acceptor substrate (1 nmol as linkage saccharides). The reaction mixtures were incubated at 37 °C for 4 h and then subjected to gel filtration using a syringe column packed with Sephadex G-25 (Superfine, Cytiva) or separated on a Superdex™ Peptide column (Cytiva) equilibrated with elution buffer (0.25 M $NH_4HCO_3$/7% 1-propanol). The incorporation of [$^{32}$P] phosphate into the acceptors was quantified[23] by determining the radioactivity in the corresponding fractions using a liquid scintillation counting detector, Tri-Carb liquid scintillation analyzer (model TRI-CARB 2900TR, PerkinElmer Life and Analytical Sciences, operated by Microsoft Windows NT, Ver. 4).

## Sulfotransferase assay

Sulfotransferase activity toward chondroitin was assessed using an established method[62] with slight modifications. Briefly, the reaction mixture contained 10 μl of resuspended beads, 50 mM imidazole-HCl (pH 6.8), 2 mM dithiothreitol, 10 μM [$^{35}$S]PAPS ($3 \times 10^5$ dpm), and chondroitin (1 nmol as GlcA). The reaction mixtures were incubated at 37 °C for 30 min and gel filtered using a disposable syringe column. Radioactivity in the flow-through fractions was measured by liquid scintillation counting.

## Pulldown assay

The cDNA fragments of the truncated forms of FAM20B or FAM20C were amplified by PCR using specific primer sets in which each primer had an *Xba*I site (Supplementary Table 8). The PCR fragment was subcloned into the *Xba*I site of the p3XFLAG-CMV™-8 expression vector (Sigma-Aldrich) to encode a fusion protein containing a 3× FLAG tag (p3XFLAG-CMV-8-FAM20B and p3XFLAG-CMV-8-FAM20C). To construct the soluble FAM20C mutants, their respective cDNA fragments were amplified by PCR with their corresponding plasmid constructs (pEF-BOS/IP-FAM20Cs, G379R, G379E, L388R, and R549W) as templates using the abovementioned universal primer set for FAM20C, with each primer containing an *Xba*I site. The fidelity of the resultant plasmid constructs (p3XFLAG-CMV-8-FAM20Cs, G379R, G379E, L388R, and R549W) was confirmed by DNA sequencing. The cDNA fragment of a truncated form of C4ST-1 that lacks the first 59 N-terminal amino acids was amplified with pCMV-C4ST-1[37] as a template using a specific primer set, each of which had a strategically distinct restriction enzyme recognition sequence (Supplementary Table 8). The fragment was inserted into a pcDNA3Ins-His expression vector[33] to encode a fusion protein with an insulin signal sequence and a 6× His sequence tag (pcDNA3Ins-His-C4ST-1).

The His-tagged and FLAG-tagged expression vectors (3.0 μg each) were co-transfected into COS-1 cells grown on 100-mm plates. Two days after transfection, 1 ml of the culture medium was collected and incubated with 10 μl Ni-NTA agarose (Qiagen, Hilden Germany) or anti-DYKDDDDK (FLAG) tag antibody-conjugated beads (Fujifilm Wako, Osaka, Japan) at 4 °C for 1 h. The beads were recovered by centrifugation and washed thrice with tris-buffered saline buffer containing 0.05% Tween 20. The proteins bound to the Ni−NTA agarose and anti-FLAG tag beads were eluted by competition with a high concentration of imidazole and an excess of free FLAG peptide, respectively. The eluates were then subjected to sodium dodecyl sulfate-polyacrylamide gel electrophoresis. The separated proteins were transferred onto a polyvinylidene fluoride membrane (Cytiva), and incubated with primary antibodies against His-tag (rabbit polyclonal, #2365, 1:1000, Cell Signaling Technology, Danvers, MA, USA) or FLAG tag (mouse monoclonal, clone M2, #F1804, 1:1000, Sigma-Aldrich), and treated with horseradish peroxidase-conjugated secondary antibodies. Enhanced chemiluminescence Select Detection Reagent (Cytiva) was used to visualize the antibody-labeled protein bands.

Pulldown experiments using Saos-2 clonal cells were conducted by formaldehyde crosslinking[65] with slight modifications. The cells of subconfluent cultures were lysed in lysis buffer (40 mM HEPES pH 7.4, 120 mM NaCl, 1 mM EDTA, 1% Triton X-100, and protease inhibitor cocktail [Nacalai Tesque, Kyoto, Japan]) with 0.1% formaldehyde. The lysate was incubated at 4 °C for 1 h, and then the crosslinking reaction was quenched by the addition of Tris-HCl, pH 7.4 (final concentration: 200 mM). The lysates were incubated with Dynabeads Protein G (Invitrogen) conjugated with anti-FAM20C antibody (rabbit polyclonal, #25395-1-AP, 1 μg; Proteintech, Rosemont, IL, USA), which was prepared using the crosslinker BS3 (#B574; Dojindo, Kumamoto, Japan). After repeated washing, the Dynabeads were resuspended in 1× LDS sample buffer containing 50 mM dithiothreitol and incubated at 99 °C for 20 min to reverse most of the formaldehyde crosslinks. The resultant sample solution containing FAM20C immunoprecipitates was analyzed by western blotting using specific antibodies against FAM20C (#25395-1-AP, 1:1000, Proteintech) and C4ST-1 (mouse monoclonal, clone L18, #sc-100868, 1:200, Santa Cruz, Dallas, TX, USA).

## Protein kinase assay

The soluble forms of FLAG-tagged FAM20B and FAM20C were purified from the conditioned media of COS-1 cells transfected with the respective FLAG-tagged expression vectors. The reaction mixture for the protein kinase assay (50 μl) contained 500 ng of purified FLAG-tagged FAM20 proteins, 50 mM HEPES (pH 7.4), 10 mM $MnCl_2$, 1 mM ATP, and an acceptor substrate (0.625 μg of recombinant human C4ST-1, #11396-H08B, Sino Biological, Beijing, China). The mixtures were incubated at 30 °C for 2 h and subjected to phosphate affinity chromatography[66] using phos-tag agarose (Fujifilm Wako), according to the manufacturer's instructions. The presence of phosphorylated C4ST-1 was analyzed by immunoblotting of the resultant phos-tag-bound materials using an anti-C4ST-1 antibody (clone L18).

## Disaccharide composition of GAGs

Isolation and purification of GAG chains from the cell cultures were performed[67]. Cells were homogenized with ice-cold acetone, extracted with ice-cold acetone three times, and air-dried thoroughly. The delipidated acetone powder was digested with actinase E (one-tenth the weight of acetone powder) in 0.1 M borate-sodium, pH 8.0, containing 10 mM $CaCl_2$ at 55 °C for 48 h. The samples were adjusted to 5% v/v in trichloroacetic acid and centrifuged. The GAG-containing materials were precipitated from the resultant supernatants by mixing with ethanol, dissolved in water, and subjected to gel filtration on a PD-10 desalting column (Cytiva) using water as an eluent. The flow-through fractions were collected and evaporated to dryness. The purified GAG fraction containing the CS and HS chains was digested with 5 mIU of chondroitinase ABC (Seikagaku), or a mixture of 0.5 mIU of heparinase (Seikagaku) and 0.5 mIU of heparitinase (Seikagaku), at 37 °C for 3 h. The digests were derivatized with the fluorophore 2-AB and then analyzed by anion-exchange HPLC using a YMC-Pack PA-G column (4.6 × 250 mm, YMC, Kyoto, Japan). A modular HPLC system (Shimadzu) was operated by LabSolutions LC/GC (Ver. 5.42, Shimadzu). The identification and quantification of the resulting disaccharides were achieved by comparison with authentic unsaturated CS and HS disaccharides (Seikagaku).

## Gel filtration chromatography of GAGs

To measure GAG chain lengths, the purified GAGs were subjected to β-elimination using $NaBH_4$/NaOH treatment and then analyzed by gel filtration chromatography on a Sephadex-200 column (10 × 300 mm, Cytiva) eluted with 0.2 M $NH_4HCO_3$ at a flow rate of 0.4 ml min$^{-1}$. Fractions were collected at 3-min intervals, lyophilized, and digested with ChABC or a mixture of heparinase and heparitinase. The digests were derivatized with 2-AB and analyzed by HPLC using a YMC-Pack PA-G column, as described above.

## Metabolic labeling

The stable HeLa cell lines were metabolically labeled with [$^{32}$P] $NaH_2PO_4$ (285.2 mCi mmol$^{-1}$) in sodium- and phosphate-free DMEM (Gibco®, Life Technologies) containing 10% dialyzed fetal bovine serum (FBS) at 37 °C for 30 h. Each cell layer was treated with 0.5 M LiOH at 4 °C for 12 h to liberate the O-linked saccharides from core proteins[68]. Each sample was subsequently neutralized and applied to a column (1 ml bed volume) of AG 50 W-X2 (H$^+$ form, Bio-Rad). Flow-through fractions containing O-linked saccharide components were pooled, neutralized with 1 mM $NH_4HCO_3$, and derivatized with 2-AB. The 2-AB derivatives were analyzed by gel filtration chromatography on a Superdex peptide column (10 × 300 mm) eluted with 0.2 M $NH_4HCO_3$ at a flow rate of 0.4 ml min$^{-1}$. [$^{32}$P] radioactivity of the pooled fractions eluted at the positions of the authentic 2AB-labeled linkage saccharide intermediates Galβ1−4Xyl(2-O-phosphate)-2AB and Galβ1−3 Galβ1-4Xyl(2-O-phosphate)-2AB was measured[25].

## Osteoblastic cultures

Saos-2 cells were cultured in a basal medium (BM; McCoy's 5a containing 15% FBS). Cells stably expressing FAM20B, FAM20C, or individual FAM20C mutants were generated by transfection with the respective pCMV expression plasmids described above. Clonal cell lines were selected and maintained in the presence of 400 μg ml$^{-1}$ or 100 μg ml$^{-1}$ G418, respectively. To assess biomineralization levels, each Saos-2 clone was plated at a density of $5 \times 10^3$ cells cm$^{-2}$, cultured in BM for the first 72 h, and then maintained for an additional 21 days in BM supplemented with 100 μg ml$^{-1}$ L-ascorbic acid 2-phosphate and 10 mM β-glycerophosphate. The cultured cells were stained with alizarin red[56]. Optical images of the stained cells were analyzed with BZ-X Analyzer (Ver. 1.4.1.1, Keyence). For the quantification of the staining intensity, the deposited dye was extracted with 5% formic acid, and the absorbance was measured at 450 nm.

To establish MC3T3-E1 cells stably overexpressing C6ST-1, the expression plasmid pCMV-C6ST-1[37] was transfected into MC3T3-E1 cells, which were cultured in the presence of 800 μg ml$^{-1}$ G418. Drug-resistant colonies were selected and propagated for subsequent experiments. Parental MC3T3-E1 cells and their stable cell lines were maintained in a growth medium (GM, α-MEM containing 10% FBS). For osteoblast differentiation, the cells were plated at a density of $5 \times 10^3$ cells cm$^{-2}$, cultured in GM for the first 72 h, and then maintained until day 28 in differentiation medium (DM; GM supplemented with 100 μg ml$^{-1}$ L-ascorbic acid 2-phosphate, 10 mM β-glycerophosphate, and 10 nM dexamethasone)[46]. ALP staining for cellular ALP activity and alizarin red staining for the measurement of mineral matrix formation were performed on days 14 and 28, respectively[46,56].

BMSCs were harvested from the femora of 8-week-old wild-type or C6ST1 transgenic male mice. To induce osteoblast differentiation, BMSCs were placed in 96-well plates at a high density ($1.5 \times 10^4$ cells/ well) and cultured for 21 days in DM[56]. ALP and alizarin red staining were performed on days 7 and 21, respectively. In some cases, BMSCs have been maintained in DM supplemented with 30 mIU ChABC from day 3 until the end of the culture period.

## Cell-to-substrate adhesion assay

Single-cell suspensions of parental MC3T3-E1 cells or their stable cell lines were labeled with a fluorescent plasma membrane dye, PKH26 (2 mM, Sigma-Aldrich). After washing in PBS to remove excess dye, the labeled cells ($1 \times 10^4$ cells) were plated on 4-well plates pre-coated with recombinant N-cadherin-Fc or cadherin-11-Fc (R&D Systems, Minneapolis, MN, USA). Cells were allowed to settle and adhere for 30–60 min in the presence of 3 mM $CaCl_2$, the non-adherent cells were gently washed away with PBS, and fluorescent images of adherent cells were obtained using an all-in-one fluorescence microscope BZ-X800 (controlled by BZ-X viewer, Ver. 01.03.02.01, Keyence). The number of fluorescent cells adhering to the substratum was counted from 10 microscopic fields. This number was used as an index of cell–substrate adhesion[46].

## Western blotting

Cell extracts from subconfluent cultures of parental Saos-2 or their clonal cells in GM were subjected to immunoblotting using antibodies against FAM20C (#25395-1-AP, 1:1000, Proteintech), FAM20B (mouse monoclonal, clone 1018512, #MAB8427, 1:250, R&D Systems), C4ST-1 (clone #L18, 1:200, Santa Cruz), and glyceraldehyde-3-phosphate dehydrogenase (GAPDH, mouse monoclonal, clone 5A12, #014-25524, 1:1000, Fujifilm Wako). Semi-confluent cultures of parental MC3T3-E1 or their stable cell lines in GM were treated for 1 h with neutralizing antibodies (5 ng ml$^{-1}$ each) for N-cadherin (GC4, #C3865, Sigma-Aldrich), cadherin-11 (16G5, #ab151446, Abcam, Cambridge, England), or an isotype control antibody, followed by the addition of CS-C polysaccharides (20 μg ml$^{-1}$, Seikagaku) for 2 h. After incubation, cell extracts were examined by immunoblotting using polyclonal

antibodies (1:1000, Cell Signaling Technology), ERK1/2 (#9102), phospho-ERK1/2 (#9101), Smad1 (#9743), phospho-Smad1/5/8 (#9511), Smad3 (#9523), and phospho-Smad3 (#9520). The respective bound antibodies were detected with sheep anti-mouse IgG HRP-linked (#NA931, 1:5000, Cytiva), donkey anti-rabbit IgG HRP-linked (#NA934, 1:10,000, Cytiva), or mouse anti-goat IgG HRP-linked (#sc-2354, 1:5000, Santa Cruz). Blotting images were obtained and analyzed with a luminescent image analyzer ImageQuant 800 (operated by IQ800 control software, Ver. 1.1.2, Amersham) and ImageQuant TL (Ver. 8.1, Amersham), respectively. Uncropped blot images are shown in a Source Data file.

### Interaction analysis

The binding of CS-C polysaccharide to N-cadherin or cadherin-11 was examined using the BIAcore J system (operated by BIAcoreJ control software, Ver. 1.1, Cytiva), with a proven track record in the molecular interaction studies for GAGs[69]. Recombinant N-cadherin-Fc or cadherin-11-Fc was immobilized on a CM5 sensor chip (Cytiva), according to the manufacturer's instructions. A series of CS-C concentrations ranging from 0.4–2.0 μM in running buffer containing 3 mM $CaCl_2$ or 3 mM EDTA was applied to the flow cells and changes in resonance units (RU) were recorded. Data were analyzed using BIA evaluation software (version 3.0; Cytiva) with a 1:1 Langmuir binding model.

### Reverse transcription-polymerase chain reaction

Total RNA was extracted from the cells using TRIzol reagent (Invitrogen). An aliquot (1 μg) of each total RNA sample was pretreated with RNase-free DNase to serve as a template for cDNA synthesis. Endpoint PCR was performed using GoTaq Hot Start polymerase (Promega). Images of the uncropped gel are shown in a Source Data file. FastStart DNA Master Plus SYBR Green I and a LightCycler ST300 (Roche Diagnostics) were used to perform quantitative real-time reverse transcription-polymerase chain reaction. Primer sequences are listed in Supplementary Table 9. The expression of each gene was normalized to that of *GAPDH* (*gapdh*).

### Assessment of bone parameters

BMD and stress-strain indexes of tibias from 16-week-old wild-type or *C6ST1* transgenic male mice were measured using peripheral quantitative CT (pQCT, XCT Research SA+, Canon Lifecare Solutions, Osaka, Japan) and dual-energy X-ray absorptiometry (DXA, DCS-600, Hitachi Aloca Medical, Tokyo, Japan). Micro-CT (LaTheta LCT-200, operated by LaTheta ver. 3.40, Hitachi Aloca Medical) analysis was also performed on tibias from wild-type or *C6ST1* transgenic mice. BMD assessment and reconstruction of 3D images of the respective tibias were conducted using LaTheta (ver. 3.40, Hitachi Aloca Medical) and VGStudio MAX 2.2 (Volume Graphics), respectively.

### Statistics and reproducibility

For all statistical plots, data are expressed as the mean ± standard deviation (s.d) of at least three independent experiments. Statistical significance was determined using Dunnett's multiple comparison method, Tukey–Kramer multiple comparison method, or unpaired Student's *t*-test, as noted in the legends to each Fig. Statistical analyses were performed with Microsoft Excel for Mac (Ver. 15.33) with a data analysis add-in, Mac statistical analysis Ver. 3.0 (Esumi). Differences with *P* values less than 0.05 were considered statistically significant. All experiments, unless otherwise indicated, were reproduced with similar results a minimum of three times, and the exact number of repetitions is provided in the legends of each Fig.

### Reporting summary

Further information on research design is available in the Nature Portfolio Reporting Summary linked to this article.

## Data availability

The authors declare that the data supporting the findings of this study are available within the paper and its Supplementary Information files, or from the corresponding author upon reasonable request. Source data are provided in this paper.

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

## Acknowledgements

This work was supported in part by the Ministry of Education, Culture, Sports, Science and Technology (MEXT)-Supported Program for the Strategic Research Foundation at Private Universities, 2012-2017 (to H.K.) and Grants-in-Aid for Scientific Research on Innovative Areas (No. 23110003 to H.K., Deciphering sugar chain-based signals regulating integrative neuronal functions), for Scientific Research B (No. 25293014, No. 16H05088, and 20H03386 to H.K.), and for Scientific Research C (No. 16K07306, and 20K06551 to T.M.) from MEXT of Japan.

## Author contributions

T.K., T.M., and H.K. designed and performed the research, analyzed the data, and wrote the paper. H.K. conceived the study. J.T. produced chemical compounds for the linkage region of saccharides.

## Competing interests

The authors declare no competing interests.
