## [Peer Review File · Nature Communications]

Altered sulfation status of FAM20C-dependent chondroitin sulfate is associated with osteosclerotic bone dysplasiaReviewers' comments:

Reviewer #1

Expert in GAG biosynthesis
(Remarks to the Author):

This is an interesting manuscript which reveals new insights into the role of FAM20C involvement in regulation of chondroitin sulfate (CS) biosynthesis, and genetic links with osteosclerotic bone dysplasia. The authors reveal for the first time that FAM20C has kinase activity for phosphorylation of the linkage tetrasaccharide, and also that it interacts directly with the CS 4-O-sulfotransferase1 (4ST1) and can regulate the level of 4-sulfation and thus the 4S/6S ratio. Known genetic mutants in FAM20C were not associated with the Xyl-kinase activity, but were associated with lack of ability of the mutant proteins to interact with and enhance C4ST1 activity. The authors contend that control of the 4S/6S ratio is crucial for physiological control of osteogenesis. Whilst they do provide a significant amount of data in Figs 4, 5 and part of Fig 6) that altering 6S biosynthesis to manipulate this ratio has effects on osteoblast differentiation and osteogenesis, this data is not a direct demonstration of the effects of altering 4ST1 activity through mutations in FAM20C. The idea of FAM20C being critical for maintaining the correct sulfation balance between 4S and 6S of CS (and loss of this balance in FAM20C-related human genetic disorders) is certainly novel, but in its current form the manuscript does not provide convincing evidence.

For example it is puzzling that mutations in FAM20C reduce its interaction with 4STs, and decrease the 4S/6S ratio, whereas this shift in ratio increases osteoblast differentiation (which is the opposite of what would be expected in the disease process. Similarly, increased 6S (through overexpression of C6ST) also decreases the 4S/6S ratio and also increases osteoblast differentiation, so this also seems counter to the authors' arguments.

Clearly the system is quite complex, and subtle changes in 4S/6S ratios and the threshold levels may well be involved. In addition, there are noted differences between mouse and human systems. This makes it even more crucial that studies are undertaken on human cells to try to resolve these issues and make clear how the results from mouse studies directly relevant to the human disease.

Major comments

The authors need to provide evidence of the effects of FAM20C mutations or at least FAM20C knockdown in relevant mouse cell lines, with analysis of altered 4S/6S ratios and concomitant alterations in osteogenesis-related outcomes which would be consistent with the disease phenotype. Although the extensive data provided on C6ST is supportive and interesting, it is insufficient to confirm the mechanisms underlying how mutant FAM20C functions.

Furthermore, the authors need to provide data on FAM20C overexpression and knockdown, and in human cells relevant to osteogenesis. Preferably this would also include examination of the activities of mutant FAM20C proteins.

Minor comments

P3, changes in CS production (~2-fold) should be described as significant, rather than dramatic.

Can the authors confirm that the changes they claim to observe in CS chain length are greater than would be expected just from the increased level of sulfation (which will increase chain molecular weight, but not length).

Many spelling errors which need to be corrected in text and figures.

Reviewer #2

Expert in FAM20C

(Remarks to the Author):

Background: Fam20C is a member of a family of proteins that shared sequence similarity (Fam20A, B, C), but were of unknown function. Fam20 proteins also displayed weak similarity to a *Drosophila* protein kinase four-jointed (Fj) (1). It has previously been shown that Fam20C is a secretory pathway protein kinase that resides in the ER and Golgi, and can also be secreted outside cells (2, 3). It has also been well-established that Fam20C is highly specific for serine residues within an SxE/pS motif found in a large number of secretory proteins (2-4). Mutations in Fam20C are causative for Raine Syndrome, a severe and often lethal bone dysplasia characterized by osteosclerosis and ectopic calcifications (5). A second member of this family, Fam20B, has been shown to play an important role in controlling proteoglycan maturation of cartilage matrix proteins, chondrocyte maturation, and perichondral bone development in zebrafish (6). Fam20B has been shown to phosphorylate a xylose residue within the tetrasaccharide linkage region of O-linked proteoglycans, thereby regulating GAG chain maturation (7, 8). The third member of this family, Fam20A, has recently been shown to allosterically potentiate Fam20C protein kinase activity, although Fam20A does not itself possess intrinsic kinase activity (9). Notably, previous studies using highly purified preparations of recombinant human Fam20C or the single ortholog present in *C. elegans* (ceFam20) have shown that Fam20C is highly specific for SxE/pS motifs within proteinaceous substrates and does not exhibit kinase activity toward artificial substrates that mimic xylose within the proteoglycan tetrasaccharide linkage (2, 4, 10). Likewise, highly purified recombinant Fam20B is active toward xylose within a minimal Gal-Xyl disaccharide and does not show activity toward SxE/pS motifs within proteinaceous substrates (7, 8, 10).

In the manuscript entitled, "Breakdown in FAM20C-mediated machinery for chondroitin sulfate biosynthesis causes osteosclerotic bone dysplasia", Koike et al presented evidence that the secretory kinase Fam20C functions to augment the 4S/6S sulfation ratio of proteoglycan chondroitin sulfate GAG chains. In this manuscript, the authors linked GAG abundance to Fam20C. In contrast to previous reports, they also showed that Fam20C may harbor intrinsic xylose kinase activity; however, this activity did not appear to be linked to Raine Syndrome. Overexpression of Fam20C in murine L cells increased the both the length and 4S/6S ratio of CS GAG chains. In contrast, these effects were not observed in a mutant cell line (sog9) that is deficient in the Golgi resident 4-O-sulfotransferase, C4ST-1. The authors used co-immunoprecipitation studies to provide evidence of a physical interaction between Fam20C and C4ST-1. They also observed an increase in relative 4-O-sulfotransferase activity when Fam20C and C4ST-1 were co-expressed. The Fam20C/C4ST-1 interaction was abrogated by Fam20C mutations causative for Raine Syndrome, suggesting that alterations in C4ST-1 activity and the 4S/6S ratio of CS GAG chains may play an important role in the etiology of Raine Syndrome patients. Finally, the

authors employed studies with the murine osteoblast cell line, MC3T3-E1, and a transgenic mouse model in which overexpression of the 6-O-sulfotransferase, C6ST-1, was used to artificially decrease the 4S/6S CS ratio without dramatically lower the overall CS abundance. C6ST-1 overexpression decreased the 4S/6S CS ratio and accelerated MC3T3-E1 osteoblast differentiation. C6ST-1 overexpression in mice led to a decreased 4S/6S CS ratio with a concomitant increase in bone mineral density that is characteristic of Raine Syndrome patients.

The hypothesis that Fam20C may play a pivotal role in regulating proteoglycan CS GAG chain abundance and 4S/6S ratio is very intriguing. Furthermore, the findings of these studies might implicate loss-of-function Fam20C mutations in the development of osteosclerotic bone dysplasia associated with severe Raine Syndrome by an additional mechanism distinct from its protein kinase activity as reported previously. However, it is the opinion of this reviewer that there are a number of major concerns regarding the work and conclusions presented in this manuscript, as outlined below:

1) The demonstration that Fam20C harbors intrinsic xylose kinase activity is unconvincing in light of previous reports in the literature to the contrary using highly purified recombinant proteins. In addition, it has also been shown that the same Raine Syndrome mutations render Fam20C kinase inactive. More importantly, both severe and hypomorphic allele Fam20C mutant protein (SxE) kinase activities correlate directly with the severity of the phenotype, strongly suggesting that it is Fam20C protein kinase activity that is critical for disease progression. Further, mutant Fam20C was incapable of secretion outside the cell, which is in direct conflict with the data shown herein. The relative efficacy in knockdown experiments should also be shown by immunoblotting, preferably with more than a single oligo, to minimize the potential for off-target effects. The significant discrepancies between the current data and prior reports should be clarified.

It has also been reported that Fam20C can physically interact with at least one other member of the family, Fam20A, suggesting that interactions between overexpressed Fam20C and other endogenous Fam20 kinases (activities) that might interfere with the clear interpretation of experimental data must be given serious consideration when conducting experiments of this kind. Given the possibility that the xylose kinase activity detected may have been an artifact caused by co-immunoprecipitation of some other secretory kinase present, it would have been more convincing had the authors used highly purified proteins, or demonstrated by immunoblotting that endogenous Fam20A/B/C were present only where expected in the samples used for xylose kinase activity analyses. Alternatively, the authors could have immunoprecipitated proteins used in kinase assays from Fam20B-deficient cells. Further, the authors demonstrate that Fam20C xylose kinase activity was not affected by Raine Syndrome mutations and suggest that it likely has no role in CS GAG chain elongation or modulating 4S/6S ratio. It is this reviewer's opinion that the early focus on Fam20C xylose kinase activity was a detraction from the story as a whole, representing primarily negative data.

2) As mentioned above, previous reports have demonstrated that Fam20C is a Golgi protein kinase that phosphorylates a wide range of secretory pathway proteins within a highly conserved SxE/pS motif. Surprisingly, the authors did not test the possibility of Fam20C might directly phosphorylate C4ST-1, which in turn could affect its transferase activity. C4ST-1 contains a well-conserved SxE motif within its C-terminus that could conceivably serve as a Fam20C target. It is unclear whether overexpression/knockdown of Fam20C could affect the expression/secretion level of C4ST-1 or other glycosyltransferases, as has been previously observed.

3) The authors should more thoroughly quantify and characterize the xylose kinase substrate, ITI. Clearly is difficult to generate these glycosylated peptides, but it is not acceptable for the authors to just assume they will get the correct glycosylation when they mixed the "bead-pulled glycosyltransferase" and UDP-sugar (pg 12). In addition, the second step of the reaction involves phosphorylation of the "xylosylated peptide (Xyl-o-TIT) by Fam20B. Previous work including the authors themselves has demonstrated that Fam20B cannot efficiently use Xyl-O-Ser as a substrate, if at all. This discrepancy should be addressed.

4) The experiments used by the authors to draw the conclusion that FAM20C can regulate GAG abundance was not convincing. Specifically, the amount of GAG was normalized to dry weight. It is unclear whether this normalization makes sense considering the possible effect of Fam20C on protein secretion (Fig 1a, 1b). Also, the relationship between GAG chain length and 4S/6S ratio were not clearly explained. Although changes in both were observed, further explanation as to whether those processes are interrelated would be of value. Both CS and HS GAGs were affected by Fam20C overexpression based on the authors' observations, but they did not further explain or comment on the effects as related to HS chains (Fig 1a, 1b, 1c, 1d). In addition, the observed effects were only statistically significant when Fam20C was overexpressed (Fig 1E). The effect of Fam20C silencing was not statistically different than the control, raising the question of physiological relevance. Finally, the change in C4 sulfation was previously reported in the authors' Fam20B paper, where they showed that Fam20B overexpression could increase C4 sulfation and Fam20B knockdown decreased C4 sulfation. This seems to be at odds with their current findings and should be discussed further.

5) In Fig 2b, it appears as though the authors radiolabeled the cells with ^{32}P , then treated the cells with base to release O-linked sugars. How do they know they are measuring Gal-Gal-Xyl(P) or Gal-Xyl(P)? It would be more convincing if the authors had structurally characterized what they actually measured, because base treatment can release many sugars, some that may also contain phosphate. Second, a better method of normalization or internal control would be more convincing than % of total radioactivity.

6) The co-immunoprecipitation between C4ST-1 and Fam20C was unconvincing as both Fam20C and C4ST were overexpressed in the cells. Co-IP might be an artifact. It would have been more convincing had at least one protein been endogenous. In addition, previous reports have indicated that the same Fam20C Raine Syndrome mutants were incapable of being secreted outside the cell. It is unclear how the authors were able to isolate these proteins from conditioned medium for use in enzymatic assays. This discrepancy should be discussed further.

7) Based on the hypothesized relationship between Fam20C and CS4T-1, the authors utilized systems in which they artificially manipulated levels of C6ST-1 to lower the 4S/6S ratio without significantly altering overall CS production. Although alteration of the 4S/6S ratio by these means clearly affected bone mineralization, it is unclear whether this might also occur by a mechanism independent of Fam20C/C4ST-1. Is it possible through these experiments to conclude that the changes initially observed in the 4S/6S ratio caused by Fam20C overexpression weren't affected by shortening of CS chains, which also was observed?

References

1. Ishikawa HO, Takeuchi H, Haltiwanger RS, & Irvine KD (2008) Four-jointed is a Golgi kinase that phosphorylates a subset of cadherin domains. *Science* 321(5887):401-404.
2. Tagliabracci VS, et al. (2012) Secreted kinase phosphorylates extracellular proteins that regulate biomineralization. *Science* 336(6085):1150-1153.
3. Ishikawa HO, Xu A, Ogura E, Manning G, & Irvine KD (2012) The Raine syndrome protein FAM20C is a Golgi kinase that phosphorylates bio-mineralization proteins. *PloS one* 7(8):e42988.
4. Tagliabracci VS, et al. (2015) A Single Kinase Generates the Majority of the Secreted Phosphoproteome. *Cell* 161(7):1619-1632.
5. Raine J, Winter RM, Davey A, & Tucker SM (1989) Unknown syndrome: microcephaly, hypoplastic nose, exophthalmos, gum hyperplasia, cleft palate, low set ears, and osteosclerosis. *Journal of medical genetics* 26(12):786-788.
6. Eames BF, et al. (2011) Mutations in *fam20b* and *xylt1* reveal that cartilage matrix controls timing of endochondral ossification by inhibiting chondrocyte maturation. *PLoS genetics* 7(8):e1002246.
7. Koike T, Izumikawa T, Tamura J, & Kitagawa H (2009) FAM20B is a kinase that phosphorylates xylose in the glycosaminoglycan-protein linkage region. *The Biochemical journal* 421(2):157-162.
8. Wen J, et al. (2014) Xylose phosphorylation functions as a molecular switch to regulate proteoglycan biosynthesis. *Proceedings of the National Academy of Sciences of the United States of America* 111(44):15723-15728.
9. Cui J, et al. (2015) A secretory kinase complex regulates extracellular protein phosphorylation. *eLife* 4:e06120.
10. Xiao J, Tagliabracci VS, Wen J, Kim SA, & Dixon JE (2013) Crystal structure of the Golgi casein kinase. *Proceedings of the National Academy of Sciences of the United States of America* 110(26):10574-10579.

Reviewer #3

Expert in osteoblastogenesis
(Remarks to the Author):

In the manuscript "Breakdown in FAM20C mediated machinery for chondroitin sulfate biosynthesis causes osteosclerotic bone dysplasia" the authors evaluated FAM20C functions and the effect of common Raine syndrome mutations in the FAM20C locus to and provide new insights into the pathogenic mechanisms of osteosclerotic bone dysplasia.

Even though the topic is very complex, they provided a very concise and comprehensible report on the probable mechanism of FAM20C.

I have the following comments:

1. The authors should provide information on the statistics for all figures.
2. Readers unfamiliar with the process of GAG synthesis would benefit from the introduction of a brief figure on the linkage tetrasaccharide synthesis and chain elongation process early in the manuscript. Also, key references on GAG sulfation are missing.

3. Please provide a reference for the statement in the discussion that the 4S/6S ratio in mice differs from humans

4) Do FAM20 mutations only affect osteoblasts or other bone cell types (osteoclasts, osteocytes) as well? How are the bone turnover marker in affected humans? Alterations in GAG sulfation are known to directly affect bone cells and signaling molecules.

**Response to the Referee's Comments**

**We are deeply grateful for their careful review and constructive comments on our manuscript.**
**The following revisions have been made, though we have spent a considerable amount of time**
**on the revisions. The cited line numbers etc are for the PDF file for your review, where**
**revisions are indicated in red.**

Reviewer #1

Expert in GAG biosynthesis

(Remarks to the Author):

This is an interesting manuscript which reveals new insights into the role of FAM20C
involvement in regulation of chondroitin sulfate (CS) biosynthesis, and genetic links with
osteosclerotic bone dysplasia. The authors reveal for the first time that FAM20C has kinase
activity for phosphorylation of the linkage tetrasaccharide, and also that it interacts directly
with the CS 4-O-sulfotransferase1 (4ST1) and can regulate the level of 4-sulfation and thus the
4S/6S ratio. Known genetic mutants in FAM20C were not associated with the Xyl-kinase
activity, but were associated with lack of ability of the mutant proteins to interact with and
enhance C4ST1 activity. The authors contend that control of the 4S/6S ratio is crucial for
physiological control of osteogenesis. Whilst they do provide a significant amount of data in
Figs 4, 5 and part of Fig 6) that altering 6S biosynthesis to manipulate this ratio has effects on
osteoblast differentiation and osteogenesis, this data is not a direct demonstration of the effects
of altering 4ST1 activity through mutations in FAM20C. The idea of FAM20C being critical
for maintaining the correct sulfation balance between 4S and 6S of CS (and loss of this balance
in FAM20C-related human genetic disorders) is certainly novel, but in its current form the
manuscript does not providing convincing evidence.

For example it is puzzling that mutations in FAM20C reduce its interaction with 4STs, and
decrease the 4S/6S ratio, whereas this shift in ratio increases osteoblast differentiation (which is
the opposite of would be expected in the disease process. Similarly, increased 6S (through
overexpression of C6ST) also decreases the 4S/6S ratio and also increases osteoblast
differentiation, so this also seems counter to the authors arguments.

Clearly the system is quite complex, and subtle changes in 4S/6S ratios and the threshold levels

may well be involved. In addition, there are noted differences between mouse and human
systems. This makes it even more crucial that studies are undertaken on human cells to try to
resolve these issues and make clarify how the results from mouse studies directly relevant to the
human disease.

6 **Major comments**

The authors need to provide evidence of the effects of FAM20C mutations or at least FAM20C
knockdown in relevant mouse cell lines, with analysis of altered 4S/6S ratios and concomitant
alterations in osteogenesis-related outcomes which would be consistent with the disease
phenotype. Although the extensive data provided on C6ST is supportive and interesting, it is
insufficient to confirm the mechanisms underlying how mutant FAM20C functions.
Furthermore, the authors need to provide data on FAM20C overexpression and knockdown, and
in human cells relevant to osteogenesis. Preferably this would also include examination of the
activities of mutant FAM20C proteins.

17 **Our response to Major comments**

According to the reviewer's comment, we examined the effects of FAM20C mutations on both
the regulation of 4S/6S ratios and the biomineralization process in human osteosarcoma cell line
Saos-2. Notably, we found that Saos-2 cells are deficient in FAM20C, and thus is an ideal
model for addressing these issues. As expected, mutant FAM20C overexpression in Saos-2 cells
led to a marked decrease in 4S/6S ratio in CS chains and enhanced biomimeralization in Saos-2
cell cultures. The results and the associated comments were added in Fig. 4, and in the text
(lines 199-216). Accompanied with these additions, we also modified the descriptions of
Abstract (lines 17-18), and methodologies (lines 332-335, 513-522, and 546-550).

27 **Minor comments**

P3, changes in CS production (~2-fold) should be described as significant, rather than dramatic.

31 **Our response to Minor comments #1**

According to the reviewer's comment, we modified the corresponding description in the text
(line 94-96).

Can the authors confirm that the changes they claim to observe in CS chain length are greater than would be expected just from the increased level of sulfation (which will increase chain molecular weight, but not length).

Our response to Minor comments #2

For evaluation of sulfation level of CS chains, sulfation degree (per CS disaccharide unit) serves as an effective index, and can be calculated as the mol% of the total sulfated disaccharides multiplied by their respective sulfate group numbers (0, 1, or 2). Notably, the sulfation degree of CS chains from Mock-transfected, and FAM20C-overexpressing HeLa cells (calculated based on data shown in Supplemental Table 1) can be estimated as 1.03 and 1.07, respectively. Since molecular weight (Mw) of a sulfate group is 80 Da, the difference in Mw between the two CS preparations (Mock & FAM20C OE) is only 3.2 Da per CS disaccharide unit. Assuming that the average chain lengths of the two CS preparations were the same, and that their unit number was 200, the difference in Mw based on the sulfate group is still only 620 Da (For reference, the average number of CS disaccharide units with a Mw of 100 kDa and sulfation degree of 1.0 is roughly estimated as 208). In contrast, as shown Fig. 1d, FAM20C overexpression resulted in a marked increase in Mw of CS chains at a level that cannot be explained by increased level of sulfation alone, compared with mock controls. Similar tendencies were also observed in L cells overexpressing FAM20C (Fig. 3b, and Supplemental Table 4). These characteristics indicated that the increase in the CS abundance in HeLa and L cells by FAM20C overexpression is mainly attributable to the increment of CS disaccharide units rather than the increased level of sulfation.

Many spelling errors which need to be corrected in text and figures.

Our response to Minor comments #3

We have carefully checked and corrected all spelling errors in the revised manuscript before resubmission.

Reviewer #2
Expert in FAM20C
(Remarks to the Author):

Background: Fam20C is a member of a family of proteins that shared sequence similarity
(Fam20A, B, C), but were of unknown function. Fam20 proteins also displayed weak
similarity to a Drosophila protein kinase four-jointed (Fj) (1). It has previously been shown that
Fam20C is a secretory pathway protein kinase that resides in the ER and Golgi, and can also be
secreted outside cells (2, 3). It has also been well-established that Fam20C is highly specific
for serine residues within an SxE/pS motif found in a large number of secretory proteins (2-4).
Mutations in Fam20C are causative for Raine Syndrome, a severe and often lethal bone
dysplasia characterized by osteosclerosis and ectopic calcifications (5). A second member of
this family, Fam20B, has been shown to play an important role in controlling proteoglycan
maturation of cartilage matrix proteins, chondrocyte maturation, and perichondral bone
development in zebrafish (6). Fam20B has been shown to phosphorylate a xylose
residue within the tetrasaccharide linkage region of O-linked proteoglycans, thereby regulating
GAG chain maturation (7, 8). The third member of this family, Fam20A, has recently been
shown to allosterically potentiate Fam20C protein kinase activity, although Fam20A does not
itself possess intrinsic kinase activity (9). Notably, previous studies using highly purified
preparations of recombinant human Fam20C or the single ortholog present in *C. elegans*
(ceFam20) have shown that Fam20C is highly specific for SxE/pS motifs within proteinaceous
substrates and does not exhibit kinase activity toward artificial substrates that mimic xylose
within the proteoglycan tetrasaccharide linkage (2, 4, 10). Likewise, highly purified
recombinant Fam20B is active toward xylose within a minimal Gal-Xyl disaccharide and does
not show activity toward SxE/pS motifs within proteinaceous substrates (7, 8, 10).

In the manuscript entitled, "Breakdown in FAM20C-mediated machinery for chondroitin sulfate
biosynthesis causes osteosclerotic bone dysplasia", Koike et al presented evidence that the
secretory kinase Fam20C functions to augment the 4S/6S sulfation ratio of proteoglycan
chondroitin sulfate GAG chains. In this manuscript, the authors linked GAG abundance to
Fam20C. In contrast to previous reports, they also showed that Fam20C may harbor intrinsic
xylose kinase activity; however, this activity did not appear to be linked to Raine
Syndrome. Overexpression of Fam20C in murine L cells increased the both the length and
4S/6S ratio of CS GAG chains. In contrast, these effects were not observed in a mutant cell
line (sog9) that is deficient in the Golgi resident 4-O-sulfotransferase, C4ST-1. The authors
used co-immunoprecipitation studies to provide evidence of a physical interaction between

Fam20C and C4ST-1. They also observed an increase in relative 4-O-sulfotransferase activity
when Fam20C and C4ST-1 were co-expressed. The Fam20C/C4ST-1 interaction was
abrogated by Fam20C mutations causative for Raine Syndrome, suggesting that alterations in
C4ST-1 activity and the 4S/6S ratio of CS GAG chains may play an important role in the
etiology of Raine Syndrome patients. Finally, the authors employed studies with the murine
osteoblast cell line, MC3T3-E1, and a transgenic mouse model in which overexpression of the
6-O-sulfotransferase, C6ST-1, was used to artificially decrease the 4S/6S CS ratio without
dramatically lower the overall CS abundance. C6ST-1 overexpression decreased the 4S/6S CS
ratio and accelerated MC3T3-E1 osteoblast differentiation. C6ST-1 overexpression in mice led
to a decreased 4S/6S CS ratio with a concomitant increase in bone mineral density that is
characteristic of Raine Syndrome patients.

The hypothesis that Fam20C may play a pivotal role in regulating proteoglycan CS GAG chain
abundance and 4S/6S ratio is very intriguing. Furthermore, the findings of these studies might
implicate loss-of-function Fam20C mutations in the development of osteosclerotic bone
dysplasia associated with severe Raine Syndrome by an additional mechanism distinct from its
protein kinase activity as reported previously. However, it is the opinion of this reviewer that
there are a number of major concerns regarding the work and conclusions presented in this
manuscript, as outlined below:

1) The demonstration that Fam20C harbors intrinsic xylose kinase activity is unconvincing in
light of previous reports in the literature to the contrary using highly purified recombinant
proteins. In addition, it has also been shown that the same Raine Syndrome mutations render
Fam20C kinase inactive. More importantly, both severe and hypomorphic allele Fam20C
mutant protein (SxE) kinase activities correlate directly with the severity of the phenotype,
strongly suggesting that it is Fam20C protein kinase activity that is critical for disease
progression. Further, mutant Fam20C was incapable of secretion outside the cell, which is in
direct conflict with the data shown herein. The relative efficacy in knockdown experiments
should also be shown by immunoblotting, preferably with more than a single oligo, to minimize
the potential for off-target effects. The significant discrepancies between the current data and
prior reports should be clarified.

It has also been reported that Fam20C can physically interact with at least one other member of
the family, Fam20A, suggesting that interactions between overexpressed Fam20C and other

endogenous Fam20 kinases (activities) that might interfere with the clear interpretation of
experimental data must be given serious consideration when conducting experiments of this
kind. Given the possibility that the xylose kinase activity detected may have been an artifact
caused by co-immunoprecipitation of some other secretory kinase present, it would have been
more convincing had the authors used highly purified proteins, or demonstrated by
immunoblotting that endogenous Fam20A/B/C were present only where expected in the
samples used for xylose kinase activity analyses. Alternatively, the authors could have
immunoprecipitated proteins used in kinase assays from Fam20B-deficient cells. Further, the
authors demonstrate that Fam20C xylose kinase activity was not affected by Raine Syndrome
mutations and suggest that it likely has no role in CS GAG chain elongation or modulating
4S/6S ratio. It is this reviewer's opinion that the early focus on Fam20C xylose kinase activity
was a detraction from the story as a whole, representing primarily negative data.

14 **Our response to Comments #1**

According to the reviewer's comment,

1) immunoblotting data to evaluate knockdown efficacy for FAM20B and FAM20C were
added in Supplementary Fig. 1c.

2) unlike previous reports, we found that FAM20C and its Raine syndrome mutants possess a
Xyl kinase (XYLK) activity. Especially, the increased production of phosphorylated GAG
linkage saccharide intermediates in FAM20C-overexpressing HeLa cells could substantially
strengthen our notion. The discrepancies between current data and the prior reports may be
attributed to the structural differences in substrates used for *in vitro* assays, because the
aglycone moieties of substrates are known to affect the GAG biosynthetic efficiency (Ref.
51 in the revised manuscript). The associated comments were added in the text (lines
103-107, 113-114, and 269-275).

3) we thought that the possibility of non-specific co-immunoprecipitation of other kinases on
the target enzyme-bound beads could be ruled out, because no kinase activity was detected
in the putative kinase-dead mutants of FAM20B (D309G) or FAM20C (D478G)
(Supplementary Table 2). The associated comments were added in the text (lines 109-113).

4) Although XYLK activity-dependent regulation of GAG abundance is not correlated with
Raine syndrome, the data for the substrate preferences of FAM20C mutants as a XYLK
provide an important insight into the FAM20B-like functional properties of FAM20C
mutants. Therefore, the XYLK-related data remain to be presented in the revised manuscript
(Fig. 2, Supplementary Figs. 2-3, and Supplementary Tables 2-3), and the associated
comments were added in the text (lines 156-158, and 280-288).

2) As mentioned above, previous reports have demonstrated that Fam20C is a Golgi protein
kinase that phosphorylates a wide range of secretory pathway proteins within a highly
conserved SxE/pS motif. Surprisingly, the authors did not test the possibility of Fam20C might
directly phosphorylate C4ST-1, which in turn could affect its transferase activity. C4ST-1
contains a well-conserved SxE motif within its C-terminus that could conceivably serve as a
Fam20C target. It is unclear whether overexpression/knockdown of Fam20C could affect the
expression/secretion level of C4ST-1 or other glycosyltransferases, as has been previously
observed.

**Our response to Comments #2**

According to the reviewer's comment, we attempted to examine whether FAM20C
phosphorylates C4ST-1. The results and associated comments were added in Supplementary Fig.
4a, and in the text (lines 183-186). Accompanied with these additions, we also modified the
descriptions of methodologies (lines 478-487).

3) The authors should more thoroughly quantify and characterize the xylose kinase substrate,
ITI. Clearly is difficult to generate these glycosylated peptides, but it is not acceptable for the
authors to just assume they will get the correct glycosylation when they mixed the "bead-pulled
glycosyltransferase" and UDP-sugar (pg 12). In addition, the second step of the reaction
involves phosphorylation of the "xylosylated peptide (Xyl-o-TIT) by Fam20B. Previous work
including the authors themselves has demonstrated that Fam20B cannot efficiently use
Xyl-O-Ser as a substrate, if at all. This discrepancy should be addressed.

**Our response to Comments #3**

According to the reviewer's comment, we attempted to characterize the ITI-substrates. However,
since the yields of the respective ITI-substrates appeared to be quite low, complete structural

analysis of ITI-substrates was so tough. Despite these uncertainties, apparent substrate
preferences of FAM20 kinases were clearly detected using the ITI-substrates; indeed,
Xyl β 1-O-ITI was a less effective substrate for both FAM20B and FAM20C, as reported
previously (Ref. 27 in the revised manuscript). Therefore, we believe the usefulness of
ITI-substrate for sensitive detection of Xyl kinase activity.

4) The experiments used by the authors to draw the conclusion that FAM20C can regulate
GAG abundance was not convincing. Specifically, the amount of GAG was normalized to dry
weight. It is unclear whether this normalization makes sense considering the possible effect of
Fam20C on protein secretion (Fig 1a, 1b). Also, the relationship between GAG chain length
and 4S/6S ratio were not clearly explained. Although changes in both were observed, further
explanation as to whether those processes are interrelated would be of value.
Both CS and HS GAGs were affected by Fam20C overexpression based on the authors'
observations, but they did not further explain or comment on the effects as related to HS chains
(Fig 1a, 1b, 1c, 1d). In addition, the observed effects were only statistically significant when
Fam20C was overexpressed (Fig 1E). The effect of Fam20C silencing was not statistically
different than the control, raising the question of physiological relevance. Finally, the change
in C4 sulfation was previously reported in the authors' Fam20B paper, where they showed that
Fam20B overexpression could increase C4 sulfation and Fam20B knockdown decreased C4
sulfation. This seems to be at odds with their current findings and should be discussed further.

22 **Our response to Comments #4**

- 1) Since the biosynthesis of GAG chains occurs in ER and Golgi apparatus,
FAM20C-dependent GAG biosynthesis is also considered to be regulated by
Golgi-localized FAM20C, but not by secreted one. After completion of GAG biosynthesis,
proteoglycans bearing GAG chains are distributed as pericellular and/or extracellular matrix
components that are easily collected by cell scraping. Therefore, dry weight of delipidated
cell pellets (or tissue homogenates) are generally utilized as a conventional way for
normalization.
2) According to the reviewer's comment, we added our interpretation of the relationship
between CS chain length and 4S/6S ratio to Fig. 7a and the legend to Fig. 7a (lines
894-898).

3) According to the reviewer's comment for FAM20C functions in HS biosynthesis, we added
a related description in the text (lines 85-87).

4) As the reviewer indicated, the effect of FAM20C knockdown in 4S/6S ratio was not
statistically significant (Fig. 1f). The 4S/6S ratio in CS chains is also affected by an
enzymatic function of chondroitin 6-*O*-sulfotransferase catalyzing CS 6-*O*-sulfation.
Therefore, we believe that such a slight decrease in 4S/6S ratio is also a physiologically
important change.

5) As the reviewer pointed out, the amount of 4-*O*-sulfated CS disaccharide unit was
substantially correlated with FAM20B expression level (Ref. 24 in the revised manuscript).
However, since FAM20B expression level also affected the amount of 6-*O*-sulfated CS
disaccharide unit, it was concluded that FAM20B expression level is inversely correlated
with the 4S/6S ratio (Ref. 24 in the revised manuscript). This property is very important for
understanding the FAM20-related Raine syndrome etiology. Therefore, we added a related
description in the text (lines 92-94, and 280-288).

5) In Fig 2b, it appears as though the authors radiolabeled the cells with ³²P, then treated the
cells with base to release O-linked sugars. How do they know they are measuring
Gal-Gal-Xyl(P) or Gal-Xyl(P)? It would be more convincing if the authors had structurally
characterized what they actually measured, because base treatment can release many sugars,
some that may also contain phosphate. Second, a better method of normalization or internal
control would be more convincing than % of total radioactivity.

**Our response to Comments #5**

We have previously established a method for identifying the phosphorylated linkage saccharide
intermediates, Gal-Gal-Xyl(2P) and Gal-Xyl(2P) (Ref 26 in the revised manuscript). Notably,
the isolated oligosaccharides from the [³²P]-metabolically labeled cells gave two major peaks
that were eluted at the positions of the abovementioned authentic linkage saccharide
intermediates on a Superdex peptide gel filtration column. Taking this advantage, we also
measured the radioactivity of the corresponding peaks derived from the respective HeLa cells

manipulating expression of FAM20 proteins in this study. That is why the ratios of the
radioactivity of two major intermediates were shown in Fig. 2b (in the original manuscript).

According to the reviewer's comment, we modified Fig. 2b, and the corresponding
descriptions in the text (lines 126-128) and in a method (Metabolic labeling, lines 507-512).
Accompanied with these correction, Fig. 1f in the original manuscript was moved to
Supplementary Fig. 3.

6) The co-immunoprecipitation between C4ST-1 and Fam20C was unconvincing as both
Fam20C and C4ST were overexpressed in the cells. Co-IP might be an artifact. It would have
been more convincing had at least one protein been endogenous. In addition, previous reports
have indicated that the same Fam20C Raine Syndrome mutants were incapable of being
secreted outside the cell. It is unclear how the authors were able to isolate these proteins from
conditioned medium for use in enzymatic assays. This discrepancy should be discussed
further.

**Our response to Comments #6**

1) According to the reviewer's comment, we conducted pulldown assay using Saos-2 clonal
cells stably overexpressing the respective FAM20 proteins. Endogenous C4ST-1 was only
able to co-immunoprecipitate with wild-type FAM20C. The results and the associated
comments were added in Supplementary Fig. 4b, and in the text (lines 208-210).
Accompanied with these additions, we also modified the descriptions of methodologies
(lines 464-477).

2) In our studies, for ease of handling, we used two kinds of soluble forms of FAM20C
proteins, which were constructed by replacing the N-terminal 42 amino acids of FAM20C
either with a cleavable insulin signal sequence and a Protein A IgG binding domain (for
enzyme source for Xyl kinase assay), or with a preprotrypsin leader sequence and 3xFLAG
tag sequence (for pulldown assay, and protein kinase assay). Thus, these types of FAM20C
mutants could be secreted and purified by their own affinity tags. To improve the
difficult-to-understand points, we modified the related descriptions in the text (lines
99-102).

7) Based on the hypothesized relationship between Fam20C and CS4T-1, the authors utilized
systems in which they artificially manipulated levels of C6ST-1 to lower the 4S/6S ratio without
significantly altering overall CS production. Although alteration of the 4S/6S ratio by these
means clearly affected bone mineralization, it is unclear whether this might also occur by a
mechanism independent of Fam20C/C4ST-1. Is it possible through these experiments to
conclude that the changes initially observed in the 4S/6S ratio caused by Fam20C
overexpression weren't affected by shortening of CS chains, which also was observed?

**Our response to Comments #7**

In contrast to osteosclerotic phenotypes in Raine syndrome patients, it has been reported that
FAM20C overexpression promotes mineralization of mouse MC3T3-E1 cell cultures, whereas
its knockdown suppresses their osteoblast differentiation (Ref 11 in the revised manuscript).
Since the decreased 4S/6S ratio observed in FAM20C mutants and by C6ST-1 overexpression
recapitulated the osteosclerotic phenotypes even in murine systems (Figs. 4-6), the wild-type
FAM20-mediated physiological functions in chain elongation of CS chains (Figs 1d, 3b and 3c)
might not affect Raine syndrome etiology. Rather, our findings emphasize the unusual
pathogenic roles of FAM20C mutants showing FAM20B-like properties that downregulate the
4S/6S ratio of CS chains. Therefore, we added a related description in the text (lines 280-311).

References

- 1. Ishikawa HO, Takeuchi H, Haltiwanger RS, & Irvine KD (2008) Four-jointed is a Golgi
kinase that phosphorylates a subset of cadherin domains. *Science* 321(5887):401-404.
- 2. Tagliabracci VS, et al. (2012) Secreted kinase phosphorylates extracellular proteins that
regulate biomineralization. *Science* 336(6085):1150-1153.
- 3. Ishikawa HO, Xu A, Ogura E, Manning G, & Irvine KD (2012) The Raine syndrome protein
FAM20C is a Golgi kinase that phosphorylates bio-mineralization proteins. *PloS one*
7(8):e42988.
- 4. Tagliabracci VS, et al. (2015) A Single Kinase Generates the Majority of the Secreted
Phosphoproteome. *Cell* 161(7):1619-1632.
- 5. Raine J, Winter RM, Davey A, & Tucker SM (1989) Unknown syndrome: microcephaly,
hypoplastic nose, exophthalmos, gum hyperplasia, cleft palate, low set ears, and osteosclerosis.
*Journal of medical genetics* 26(12):786-788.
- 6. Eames BF, et al. (2011) Mutations in fam20b and xylt1 reveal that cartilage matrix controls

timing of endochondral ossification by inhibiting chondrocyte maturation. PLoS genetics
7(8):e1002246.
7. Koike T, Izumikawa T, Tamura J, & Kitagawa H (2009) FAM20B is a kinase that
phosphorylates xylose in the glycosaminoglycan-protein linkage region. The Biochemical
journal 421(2):157-162.
8. Wen J, et al. (2014) Xylose phosphorylation functions as a molecular switch to regulate
proteoglycan biosynthesis. Proceedings of the National Academy of Sciences of the United
States of America 111(44):15723-15728.
9. Cui J, et al. (2015) A secretory kinase complex regulates extracellular protein
phosphorylation. eLife 4:e06120.
10. Xiao J, Tagliabracci VS, Wen J, Kim SA, & Dixon JE (2013) Crystal structure of the Golgi
casein kinase. Proceedings of the National Academy of Sciences of the United States of
America 110(26):10574-10579.

Reviewer #3

Expert in osteoblastogenesis

(Remarks to the Author):

In the manuscript "Breakdown in FAM20C mediated machinery for chondroitin sulfate
biosynthesis causes osteosclerotic bone dysplasia" the authors evaluated FAM20C functions
and the effect of common Raine syndrome mutations in the FAM20C locus to and provide new
insights into the pathogenic mechanisms of osteosclerotic bone dysplasia.

Even though the topic is very complex, they provided a very concise and comprehensible report
on the probable mechanism of FAM20C.

I have the following comments:

1. The authors should provide information on the statistics for all figures.

**Our response to Comments #1**

According to the reviewer's comment, we provided statistical information in all figures, for
which statistical analyses were performed.

2. Readers unfamiliar with the process of GAG synthesis would benefit from the introduction of a brief figure on the linkage tetrasaccharide synthesis and chain elongation process early in the manuscript. Also, key references on GAG sulfation are missing.

Our response to Comments #2

According to the reviewer’s comment, we added a brief figure for GAG biosynthesis in Fig. 1a in the revised manuscript, and added key references on GAG sulfation (Ref. 21 in the revised manuscript) in the text (lines 172-175).

3. Please provide a reference for the statement in the discussion that the 4S/6S ratio in mice differs from humans

Our response to Comments #3

Unfortunately, there are no reports mentioning the difference in 4S/6S ratios between humans and mice. However, the differences are very clear even when comparing the 4S/6S ratios of cells used in this study. In fact, the 4S/6S ratios of mouse-derived cells (MC3T3-E1 and fibroblastic L cells) are around 20-50 (Fig.5a, and Supplementary Table 4), but those of human-derived cells (HeLa and Saos-2 cells) are around less than 10 (Figs. 1f and 4c). In the revised manuscript, we added the associated comments in the text (lines 221-225, and 289-291).

4) Do FAM20 mutations only affect osteoblasts or other bone cell types (osteoclasts, osteocytes) as well? How are the bone turnover marker in affected humans? Alterations in GAG sulfation are known to directly affect bone cells and signaling molecules.

Our response to Comments #4

1) At present, we do not have any answer to the reviewer’s questions about the effects of FAM20C mutations on other bone cell types, and bone turnover. As stated in Discussion section (lines 315-317), further studies focusing on FAM20 family molecules are needed for a comprehensive understanding of the biosynthetic machineries that fine tune GAG chains, and for a complete picture of Raine syndrome etiology.

2) We have previously reported that a highly sulfated CS subtype, CS-E, fine-tunes osteoblast
differentiation via several intracellular signaling pathways by binding to N-cadherin and
cadherin-11 (Ref. 47 in the revised manuscript). In this study, we found that CS-C, a typical
6-*O*-sulfated CS preparation, could also stimulate the cadherin-mediated onset of osteoblast
differentiation. Therefore, we believe that several descriptions in the text (lines 226-244), and in
Discussion section (lines 301-311) are one of answers to the reviewer's question about the
regulatory roles of GAG sulfation in bone cells, and in the associated signaling molecules.
All other corrected or modified words, phrases, sentences, and recalculated values have been
shown in red in the attached file for review.

REVIEWERS' COMMENTS

Reviewer #1 (Remarks to the Author):

This is an interesting manuscript which reveals new insights into the role of FAM20C involvement in regulation of chondroitin sulfate (CS) biosynthesis, and genetic links with osteosclerotic bone dysplasia. The authors reveal for the first time that FAM20C has kinase activity for phosphorylation of the linkage tetrasaccharide, and also that it interacts directly with the CS 4-O-sulfotransferase1 (4ST1) and can regulate the level of 4-sulfation and thus the 4S/6S ratio. Known genetic mutants in FAM20C were not associated with the Xyl-kinase activity, but were associated with lack of ability of the mutant proteins to interact with and enhance C4ST1 activity. The authors contend that control of the 4S/6S ratio is crucial for physiological control of osteogenesis. The idea of FAM20C being critical for maintaining the correct sulfation balance between 4S and 6S of CS (and loss of this balance in FAM20C-related human genetic disorders) is certainly novel and important. This extensively revised manuscript with additional data has addressed previous concerns and is now suitable for publication

Reviewer #3 (Remarks to the Author):

The authors have addressed all of my comments.